# Structural state recognition facilitates tip tracking of EB1 at growing microtubule ends

Taylor A Reid[1], Courtney Coombes[1], Soumya Mukherjee[1], Rebecca R Goldblum[2,3], Kyle White[1], Sneha Parmar[1], Mark McClellan[1], Marija Zanic[4], Naomi Courtemanche[1], Melissa K Gardner[1]*

[1]Department of Genetics, Cell Biology, and Development, University of Minnesota, Minneapolis, United States; [2]Medical Scientist Training Program, University of Minnesota, Minneapolis, United States; [3]Department of Biochemistry, Molecular Biology, and Biophysics, University of Minnesota, Minneapolis, United States; [4]Department of Cell and Developmental Biology, Vanderbilt University, Nashville, United States

**Abstract** The microtubule binding protein EB1 specifically targets the growing ends of microtubules in cells, where EB1 facilitates the interactions of cellular proteins with microtubule plus-ends. Microtubule end targeting of EB1 has been attributed to high-affinity binding of EB1 to GTP-tubulin that is present at growing microtubule ends. However, our 3D single-molecule diffusion simulations predicted a ~ 6000% increase in EB1 arrivals to open, tapered microtubule tip structures relative to closed lattice conformations. Using quantitative fluorescence, single-molecule, and electron microscopy experiments, we found that the binding of EB1 onto opened, structurally disrupted microtubules was dramatically increased relative to closed, intact microtubules, regardless of hydrolysis state. Correspondingly, in cells, the blunting of growing microtubule plus-ends by Vinblastine was correlated with reduced EB1 targeting. Together, our results suggest that microtubule structural recognition, based on a fundamental diffusion-limited binding model, facilitates the tip tracking of EB1 at growing microtubule ends.
DOI: https://doi.org/10.7554/eLife.48117.001

*For correspondence:
klei0091@umn.edu

Competing interests: The authors declare that no competing interests exist.

## Introduction

Microtubules are long, thin polymers that mechanically contribute to cell morphology, act as a track for molecular motor-based transport within the cell, and serve as a platform for binding of microtubule-associated proteins (*Howard and Hyman, 2003*; *Mitchison and Kirschner, 1984*; *Ross et al., 2008*). Microtubules are composed of $\alpha\beta$ tubulin heterodimers stacked end-to-end into 'protofilaments'. Typically, a microtubule is composed of 13 laterally associated protofilaments (*Tilney et al., 1973*). Each tubulin heterodimer contains an exchangeable nucleotide site on the β-tubulin subunit, and individual tubulin heterodimers polymerize onto the plus-end of the microtubule with the β-tubulin subunit bound to a GTP nucleotide. After integration into the microtubule, the β-tubulin-bound nucleotide then stochastically undergoes hydrolysis and converts via GDP-Pi to GDP. GDP-bound tubulin is less stable in the microtubule lattice than GTP-bound tubulin, but the microtubule remains intact and continues to grow at its plus-end due to the continued addition of GTP-bound tubulin, which forms a 'GTP cap' at the growing plus-end of the microtubule (*Desai and Mitchison, 1997*; *Mitchison and Kirschner, 1984*).

Recent literature suggests that microtubule structure may be more complex than previously considered, especially at the microtubule plus-end. A perfectly intact, closed microtubule lattice with

'blunt' ends is defined by a regular arrangement of tubulin dimers into a thirteen protofilament tube, with all equal-length protofilaments terminating at the microtubule ends. In contrast, microtubule plus-ends have been observed by cryo-electron microscopy to have open, sheet-like or tapered conformations, which diverge greatly from a closed tube conformation (*Chrétien et al., 1995*; *Guesdon et al., 2016*; *Manka and Moores, 2018a*). These findings have also been supported by quantitative analysis of fluorescence images (*Coombes et al., 2013*). This type of tapered, gently curved tip structure likely plays a role in the binding of Doublecortin and other proteins to microtubule plus-ends (*Bechstedt and Brouhard, 2012*; *Bechstedt et al., 2014*); reviewed in *Brouhard and Rice (2014)*. Further, it has recently been reported that lattice damage and tubulin turnover can occur on the microtubule lattice itself, leading to irregularities along the length of a dynamic microtubule (*Schaedel et al., 2015*).

The microtubule tip tracking protein EB1 is a highly conserved protein that autonomously tracks the growing plus-ends of microtubules (*Bieling et al., 2007*; *Dixit et al., 2009*; *Morrison et al., 1998*). At the plus-end, EB1 recruits many other +TIP family proteins that have little to no native affinity for microtubules but that must localize to microtubule plus ends to perform their functions (*Bieling et al., 2007*; *Dixit et al., 2009*; *Lansbergen and Akhmanova, 2006*). High-affinity binding of EB1 to the GTP-cap at growing microtubule plus-ends may contribute to its plus-end localization (*Maurer et al., 2011*; *Zanic et al., 2009*). The increased affinity of EB1 for GTP tubulin relative to GDP tubulin has been demonstrated through the use of GTP analogs, most commonly GMPCPP and GTP-γ-S. In both cases, there was an increase in overall EB1 binding to the GTP-analog-bound microtubules as compared to GDP-microtubules (*Maurer et al., 2011*; *Zanic et al., 2009*). However, the mechanism for how EB1 rapidly and efficiently targets to growing microtubule plus-ends, thus allowing for robust tip tracking, remains unknown.

In this work, we perfomed 3D single-molecule diffusion simulations, which predicted a ~ 6000% increase in EB1 arrivals to open, tapered microtubule tip structures relative to closed lattice conformations. Using quantitative fluorescence, single-molecule, and electron micrcopy experiments, we found that the binding of EB1 onto opened, structurally disrupted microtubules was dramatically increased relative to closed, intact microtubules, regardless of hydrolysis state. Further, we converted growing microtubule ends in LLC-Pk1 cells from a tapered to a blunt configuration, and observed a dose-dependent reduction in EB1 targeting. Together, our results suggest that microtubule structural recognition, based on a simple diffusion-limited binding model, facilitates EB1 tip tracking at growing microtubule plus-ends.

## Results

### At intermediate salt concentrations, EB1 preferentially binds to GMPCPP microtubule end structures

Previous in vitro studies have shown that while EB1 uniformly coats GTP-analog (GMPCPP) microtubules in the absence of added salt, the addition of KCl increasingly drives EB1 off of the GMPCPP lattice (*Dixit et al., 2009*; *Maurer et al., 2011*; *Zanic et al., 2009*). Thus, we used an intermediate KCl concentration to explore the localization of EB1-GFP on reconstituted microtubules under relatively weak EB1 binding conditions (*Figure 1A*, 30 mM KCl). We reasoned that under these conditions, any localized binding of EB1-GFP to GMPCPP microtubules could reveal an additional layer of regulation for EB1 binding, beyond exclusively tubulin subunit nucleotide state. Interestingly, we observed that, at 30 mM KCl, EB1 was often localized to GMPCPP microtubule ends (*Figure 1B*, green).

To detect whether there could be an underlying structure of the GMPCPP microtubule ends that would predispose them to EB1-GFP binding, we collected intensity line-scans of green EB1-GFP and red (rhodamine) GMPCPP microtubule fluorescence along the microtubules' lengths to create average intensity profiles over all microtubules. The line-scans for 64 microtubules of similar length were rebinned to the mean microtubule length (see *Gardner et al., 2008*) and Materials and methods), and the ensemble average across all 64 microtubules was plotted for both the red and green channels (*Figure 1C*). Prior to averaging, the line-scans in both the red and green channels for each microtubule were oriented such that the microtubule end with the brighter intensity of green EB1-

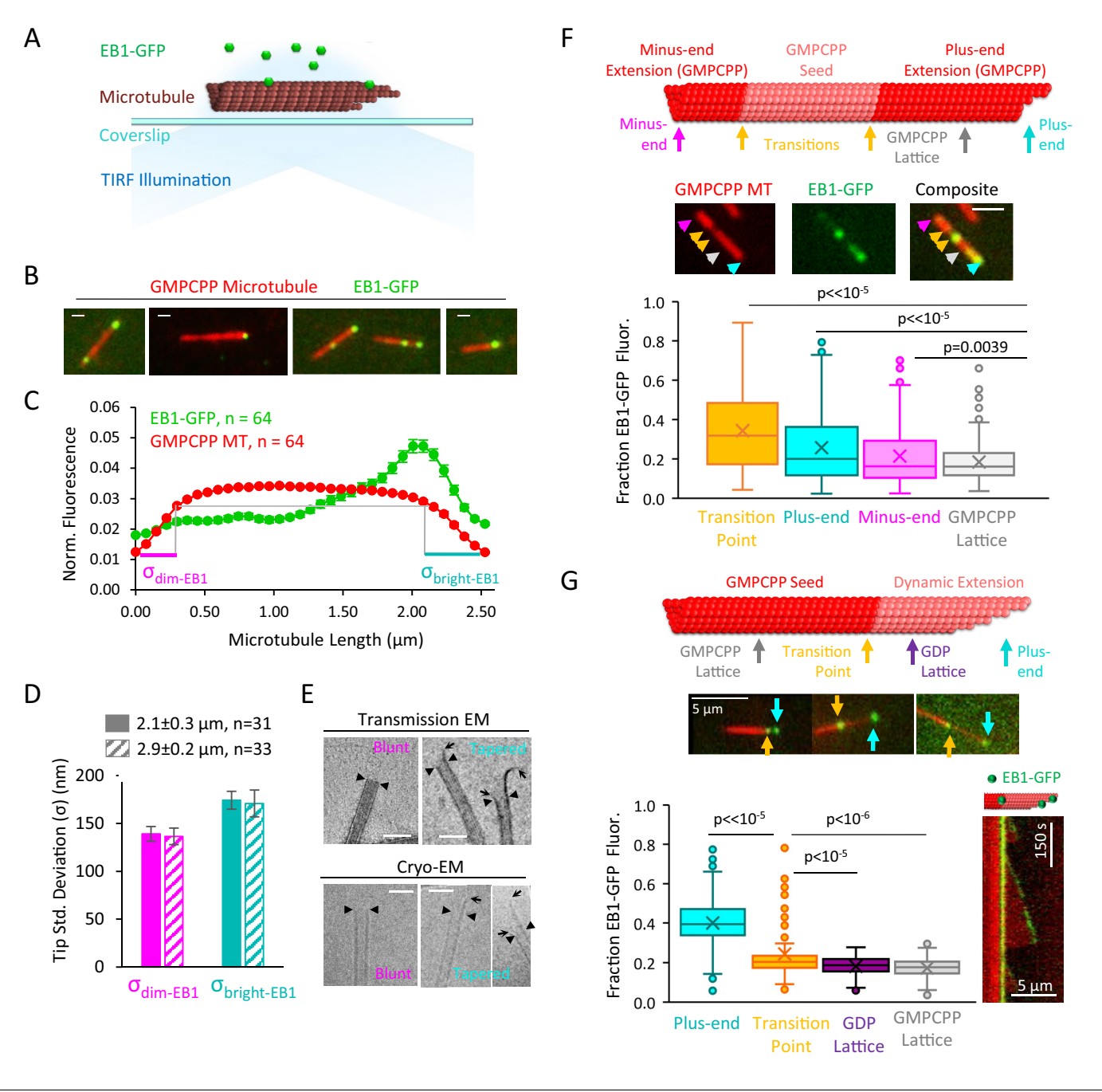

**Figure 1.** EB1-GFP recognizes structural disruptions and tapered ends on GMPCPP microtubules. (**A**) Schematic of EB1-GFP TIRF experiment. (**B**) Representative images of EB1-GFP (green) on rhodamine-labeled GMPCPP microtubules (red), showing preferential end localization. Scale bars: 0.4 μm. (**C**) Super-averaged intensity profile of 64 microtubules of length 2.5 ± 0.48 μm (mean ± SD) with EB1-GFP. Individual microtubules within the averaged line scan were aligned with brighter EB1-GFP fluorescence on the right. Error bars show standard error at each point. (**D**) Bar graph with fitted tip standard deviations for microtubules of length 2.1 ± 0.3 μm (solid bars) and 2.9 ± 0.2 μm (hashed bars) (curve fits shown in *Figure 1—figure supplement 1A*). Low tip standard deviation corresponds to blunt microtubule ends (left), high tip standard deviation correspond to ends with more variable protofilament lengths (right). Error bars: SE calculated from 95% confidence intervals. (**E**) Electron microscopy images of blunt and tapered GMPCPP microtubules ends. Closed triangles indicate the most distal portion of the microtubule with a complete tubular lattice. Open arrows indicate the extrema of protruding protofilaments. Scale bars: 50 nm. (**F**) Top: Schematic of polarity-marked GMPCPP microtubule layout and reference positions for analysis. Center: Representative TIRF images of EB1-GFP on GMPCPP microtubules, colored arrows indicate the corresponding points used for analysis. Scale bars: 3 μm. Bottom: Box and whisker plot of the fraction of total EB1 fluorescence located at each position. The two transition points were averaged to provide a single value. Plus-ends, minus-ends, and transition points all showed significant differences as compared to lattice

*Figure 1 continued on next page*

*Figure 1 continued*

binding region (n = 309 microtubules). (G) Top: Schematic of bright GMPCPP seed with dim dynamic microtubule extension, with reference positions for analysis. Center: Representative TIRF images of EB1-GFP on GMPCPP microtubules, colored arrows indicate the corresponding points used for analysis. Scale bars: 5 μm. Bottom-left: Box and whisker plot of the fraction of total EB1 fluorescence located at each position. Plus-ends and transition points both showed significant differences as compared to lattice binding region (n = 134 microtubules). Bottom-right: Representative kymograph of EB1-GFP at transition point and plus-end during dynamic microtubule growth.

DOI: https://doi.org/10.7554/eLife.48117.002

The following figure supplement is available for figure 1:

**Figure supplement 1.** Additional data to support *Figure 1*.

DOI: https://doi.org/10.7554/eLife.48117.003

GFP fluorescence was on the right side, with the dimmer EB1-GFP intensity on the left side (*Figure 1C*, green, ensemble average over n = 64 microtubules) (*Coombes et al., 2016*).

Similar to our qualitative observations, an increase in EB1-GFP intensity was observed at the brighter EB1-GFP microtubule end as compared to the center region and the dimmer EB1-GFP end of the microtubule (*Figure 1C*). Strikingly, we noted that by aligning the brighter EB1-GFP intensity on the right side of the line scan, this resulted in a qualitative difference in the underlying microtubule intensity profile at the microtubule ends (*Figure 1C*, red, ensemble average over n = 64 microtubules). Here, the red microtubule fluorescence on the right (brighter EB1-GFP) side of the line-scan plot appeared to drop off more slowly to background as compared to the left side of the line-scan plot (*Figure 1C*, compare teal vs magenta line lengths).

To quantify this observation, we fit a Gaussian error-function to the red fluorescence intensity drop-off at each microtubule end, as previously described (*Coombes et al., 2013*; *Demchouk et al., 2011*). This fitting process allowed us to estimate a 'tip standard deviation', which is a measure of tip tapering that arises as a result of protofilament length variability at the microtubule ends. To ensure that the tip fitting was not biased by rebinning a large range of microtubule lengths into one standard length, we subsampled two groups of microtubules (length standard deviation ≤0.3 μm in each case). Importantly, regardless of microtubule length, we found that the microtubule end with a brighter EB1-GFP signal had a larger average tip standard deviation, and thus more tip tapering due to protofilament length variation, than the opposite, dimmer EB1-GFP signal end, of the microtubule (*Figure 1D*; 2.1 ± 0.3 μm group (mean ± SD): p=0.00017, Z-statistic = 3.79; 2.9 ± 0.2 μm group (mean ± SD): p=0.013, Z-statistic = 2.47; Fit curves in *Figure 1—figure supplement 1A,B*). We then used both Transmission Electron Microscopy (TEM) and Cryo-Electron Microscopy to verify that tapered and blunt ends were indeed present in the GMPCPP microtubule population (*Figure 1E*), similar to previous observations (*Atherton et al., 2018*).

We then asked whether the inclusion of specific structural disruption points on the GMPCPP microtubules would alter EB1-GFP localization. Thus, polarity-labeled GMPCPP microtubule seeds were generated by growing bright-red GMPCPP extensions from dim-red GMPCPP seeds (*Figure 1F*, top). These two-color microtubules allowed for identification of the microtubule plus-end as the end with the longer bright-red extension (*Figure 1F*, top, teal). However, importantly, the polarity-labeled seeds also generated transition points between the original GMPCPP seeds and the GMPCPP extensions, which likely caused lattice discontinuities, defects, and holes at each transition point (*Figure 1F*, top, yellow). Thus, we analyzed the brightness of EB1-GFP at five positions along the GMPCPP microtubules (*Figure 1F*, top): the minus end tip (magenta), the two transition points (yellow), the plus-end tip (cyan), and the GMPCPP lattice (gray; for consistency and to minimize overlap with other positions, the lattice position used was always halfway between the plus-end and the nearest transition point). At each of the five positions, we summed the total green EB1-GFP intensity over a 9-pixel box, centered at the defined position, and then reported the fraction of EB1-GFP intensity at each position (=box intensity/summed intensity over all five boxes). For the transition point intensity, we reported an average of the two transition point box intensities for each microtubule. We found that the average transition point (p$\ll$10$^{-5}$, t-test), plus-end (p<10$^{-8}$, t-test), and minus end (p=0.0039, t-test) of the GMPCPP microtubules all showed a significantly higher fraction of EB1-GFP fluorescence than the GMPCPP lattice position (*Figure 1G*; n = 309 microtubules).

Finally, to determine whether specific structural disruption points on the microtubule lattice could lead to EB1 binding even in the presence of dynamic microtubules, dim-red dynamic (GTP)

microtubules were grown from bright-red GMPCPP seeds (*Figure 1G* top) in the presence of 100 nM EB1-GFP and 55 mM KCl (*Gell et al., 2010*). We collected independent, individual microtubule images (*Figure 1G* middle), and analyzed the brightness of EB1-GFP within a 9-pixel box at four positions along the microtubules (*Figure 1G*, top and middle): the growing GTP-tubulin plus-end tip (cyan), the GDP lattice (purple), the transition point between the GMPCPP seed and the dynamic GDP microtubule extension (yellow), and the GMPCPP lattice (gray; note that for consistency and to minimize overlap with other positions, the GMPCPP and GDP lattice positions used were always half-way between the microtubule end and the nearest transition point). We found that, as would be expected, EB1-GFP intensity was ~2.3 fold (230%) higher at growing microtubule plus-ends as compared to the overall average intensity on the GDP lattice and GMPCPP seeds (*Figure 1G* bottom-left, cyan). However, consistent with our GMPCPP microtubule results, the EB1-GFP fluorescence at the GMPCPP/GDP transition point was 38% higher than the average intensity on the GDP lattice and GMPCPP seeds (*Figure 1G* bottom-left; $p<10^{-5}$ vs GDP lattice, $p<10^{-6}$ vs GMPCPP seeds, n = 134 microtubules). Further, in kymographs of dynamic microtubules, targeting of EB1-GFP to the GMPCPP/GDP transition occasionally persisted throughout entire microtubule growth and shortening events (*Figure 1G* bottom-right).

## EB1 preferentially binds to disrupted-structure microtubules on GMPCPP, GTPγS, and GDP microtubule populations

Taken together, our GMPCPP and dynamic microtubule results suggested that EB1 may preferentially bind to regions of microtubules that have extended, tapered tip structures, or a discontinuous tubular lattice. To test this idea using microtubule populations with various tubulin-bound nucleotides, we used a previously published method to create pools of microtubules with common nucleotide states, but with varying degrees of lattice structural integrity (*Reid et al., 2017*). These protocols allowed for generation of 'closed' microtubules with relatively intact, closed lattice structures (*Figure 2A*, left), and 'disrupted-structure' microtubules with lattice structures that had increased frequencies of defects, gaps in the lattice, and open sheet conformations (*Figure 2A*, right, red subunits at structural disruptions). These 'closed' and 'disrupted-structure' populations were described for the three most commonly used in vitro microtubule nucleotides (GDP, GMPPCP, and GTPγS (*Figure 2B*)) (*Reid et al., 2017*), and so we used these populations as a tool to reveal potential differences in EB1 binding based on microtubule structure. Here, closed and disrupted-structure populations were made for GDP and GTPγS populations via taxol treatments, which created breaks, holes, and tapering in the microtubules, followed by differing storage conditions to alter the degree of overnight repair (*Figure 2B*) (see (*Reid et al., 2017*) and Materials and methods). In contrast, disrupted-structure GMPCPP microtubule populations were generated via $Ca^{2+}$ treatment (*Figure 2B*, center) (see (*Reid et al., 2017*) and Materials and methods). Careful quantification that demonstrated a statistically significant structural disruption in the 'disrupted-structure' populations as compared to the respective closed microtubule population in each case was previously performed using both fluorescent tubulin repair assays, and electron microscopy (*Reid et al., 2017*). We note that while the preparation protocols for microtubules using the three different nucleotides were distinct, each of these protocols reflected commonly used methods for producing stabilized in vitro microtubules, and so no extreme or unusual conditions or preparation techniques were employed in preparing the microtubules.

Thus, closed and disrupted-structure microtubule populations were generated for each nucleotide type, and were separately introduced into imaging chambers. Time was allowed for the microtubules to adhere to the coverslip surface, and then a solution of EB1-GFP and imaging buffer was introduced into the chambers (see Materials and methods). After allowing time for EB1-GFP binding to reach steady-state (20–30 min), the microtubules with bound EB1-GFP were imaged using TIRF microscopy. We then reported the average EB1-GFP fluorescence intensity over green background for each microtubule, which was then normalized to the average Rhodamine-tubulin fluorescence intensity over red background for each microtubule, resulting in an 'EB1-GFP/tubulin intensity ratio' that was reported for each individual microtubule in the analysis (see Materials and methods).

For the taxol-stabilized GDP microtubules, we observed a marked increase in the EB1-GFP/tubulin intensity ratio for the disrupted-structure pool of microtubules as compared to the closed microtubules (*Figure 2C*, top). Upon quantification, the average EB1-GFP/tubulin intensity ratio in the disrupted-structure microtubule pool was ~98% higher than the average EB1-GFP/tubulin intensity

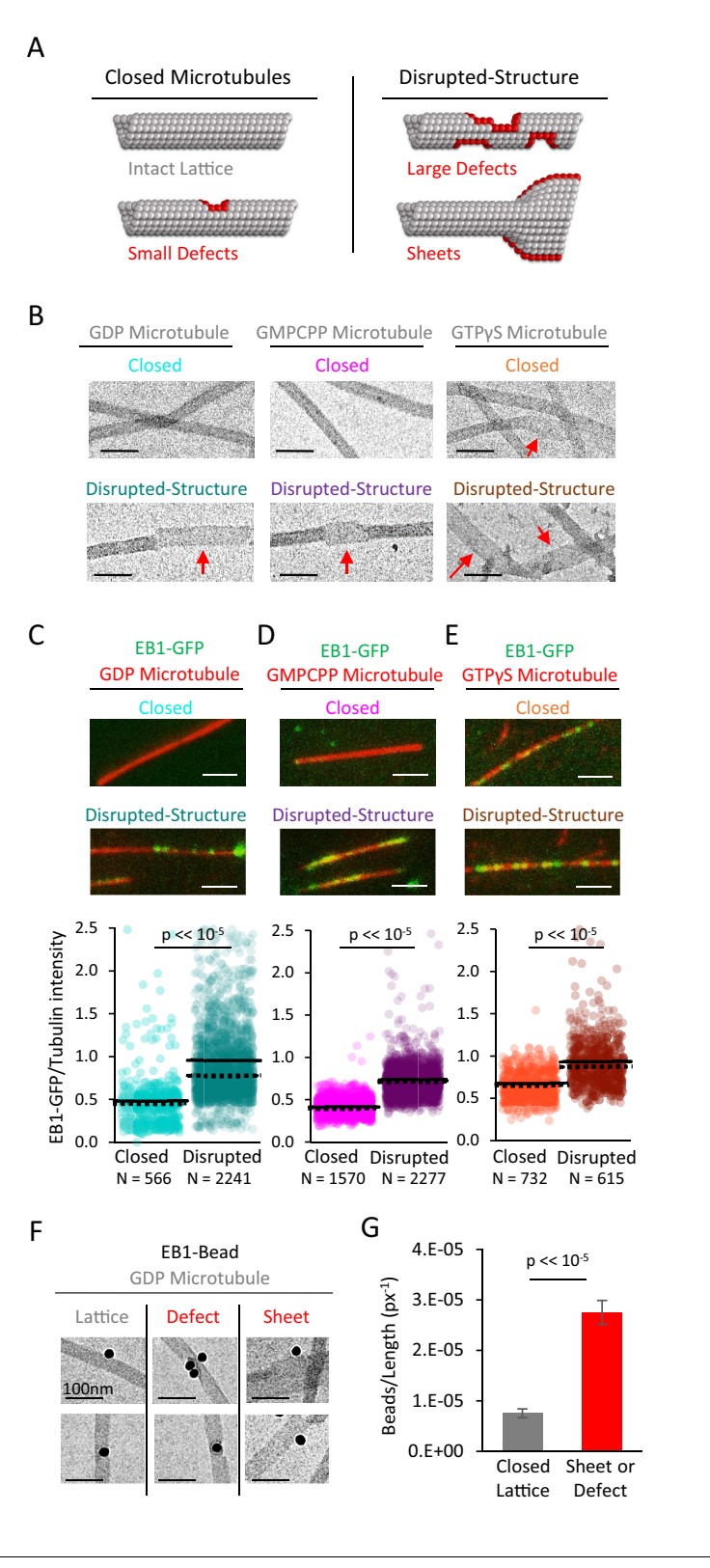

**Figure 2.** EB1-GFP shows preferential binding to disrupted-structure microtubules, regardless of nucleotide state. (**A**) Illustration of the two categories of microtubules used in this analysis. Left: 'Closed' microtubules with predominantly intact, closed lattice structures. Right: 'Disrupted-structure' microtubules have larger and more frequent gaps, defects, and sheet-like regions. Stylized red tubulin used to highlight defects. (**B**) Electron

*Figure 2 continued on next page*

*Figure 2 continued*
microscopy images of closed and disrupted-structure microtubules for GDP microtubules (left), GMPCPP microtubules (center), and GTPγS microtubules (right). Red arrows indicate structural disruptions, scale bars 50 nm. (C–E) Top: EB1-GFP (green) on microtubules (red) for closed and disrupted-structure conditions (scale bars 2 μm). Bottom: Cloud plots of EB1-GFP/Tubulin intensity ratio for individual microtubules. Each semi-transparent circle is the data point from a single microtubule. Solid lines are the average EB1-GFP/Tubulin binding ratio for each population, dotted lines are the median EB1-GFP/Tubulin binding ratio for each population. (F) Electron microscopy of EB1 conjugated to gold beads on GDP microtubules. Example images of EB1-beads bound on the microtubule lattice (left), at a lattice defect (center), and on a sheet (right). (G) Frequency of EB1-beads bound to the microtubule lattice compared to EB1-beads bound at either a defect or on a sheet, normalized to observed microtubule length in each category. Non-normalized values yield a ~ 1.7 fold increase in binding to defects and edges relative to a closed lattice (n = 74 on lattice, n = 128 on defect or sheet). Error bars SEM.
DOI: https://doi.org/10.7554/eLife.48117.004

ratio in the closed microtubule pool (*Figure 2C*, bottom; $p \ll 10^{-5}$, t-test). GMPCPP microtubules from the disrupted-structure pool also showed an increase in the average EB1-GFP/tubulin intensity ratio (*Figure 2D*, top), which was ~75% higher than the average EB1-GFP/tubulin intensity ratio in the closed microtubules (*Figure 2D*, bottom; $p \ll 10^{-5}$, t-test). Finally, the average EB1-GFP/tubulin intensity ratio was higher on disrupted-structure GTPγS microtubules as compared to closed GTPγS microtubules, with a quantitative increase of ~50% (*Figure 2E*; $p \ll 10^{-5}$, t-test).

## EB1 preferentially binds to disrupted-structure microtubules in electron microscopy experiments

To directly test whether EB1 preferentially binds to disrupted microtubule lattice structures on individual GDP microtubules, we used EB1 conjugated to gold beads, and examined the binding of these beads onto pre-stabilized microtubules using electron microscopy (*Figure 2F*). For each microtubule-bound bead, the microtubule-binding region was classified either as a complete lattice, or as a defect/sheet, and, further, the total available microtubule length for each classification was determined. Overall, we observed more EB1-gold beads that were bound to defects or sheets than to complete lattice regions (n = 74 on lattice, n = 128 on defect or sheet), regardless of total available length for each microtubule classification. However, when we normalized the frequency of bead location to the total available microtubule length for each classification, we found that the EB1-gold beads were nearly four-fold more likely to bind at defects or sheets than to a complete microtubule lattice (*Figure 2G*; $p < 2.3 \times 10^{-54}$, Chi-squared test).

## Single-molecule diffusion simulations predict a dramatic increase in EB1 on-rate to tapered microtubule structures

Taken together, our experimental results suggested that EB1 preferentially binds to regions of microtubules that have extended, tapered tip structures, or to a discontinuous tubular lattice. However, the mechanism for this preference remained unclear. Therefore, we developed a single-molecule diffusion-based computational simulation to explore a mechanistic explanation for our results.

For simplicity, we included three fundamental rules in the simulation: (1) EB1 molecules diffused in three dimensions with translational and rotational diffusion coefficients that depended on EB1 protein size (*Figures 3A*, 1; see Materials and methods) (*Castle and Odde, 2013*; *Mirtich, 1998*); (2) EB1 molecules could not pass through a microtubule, but rather would collide and then diffuse in another direction (*Figures 3A*,2); and (3) the conformation of EB1, and its binding pocket within the microtubule lattice, were representative of their published molecular structures (*Figure 3A*,3). To develop an approximation for the EB1 binding pocket within the microtubule, we used previously published Cryo-EM structural data as our guide (*Figure 3B*, left; source data from *Zhang et al., 2015*), and from this data, we created an approximation of the tubulin heterodimer shape (*Figure 3B*, center), with careful consideration for EB1's binding pocket, located between four tubulin dimers. EB1's microtubule binding domain shape was similarly modeled from previously published Cryo-EM structures (*Zhang et al., 2015*), and the modeled shape was then verified to fit into our modeled 4-tubulin binding pocket in the correct orientation (3B, right). We note that each EB1 molecule could bind only one unique tubulin interface, and in only one unique orientation.

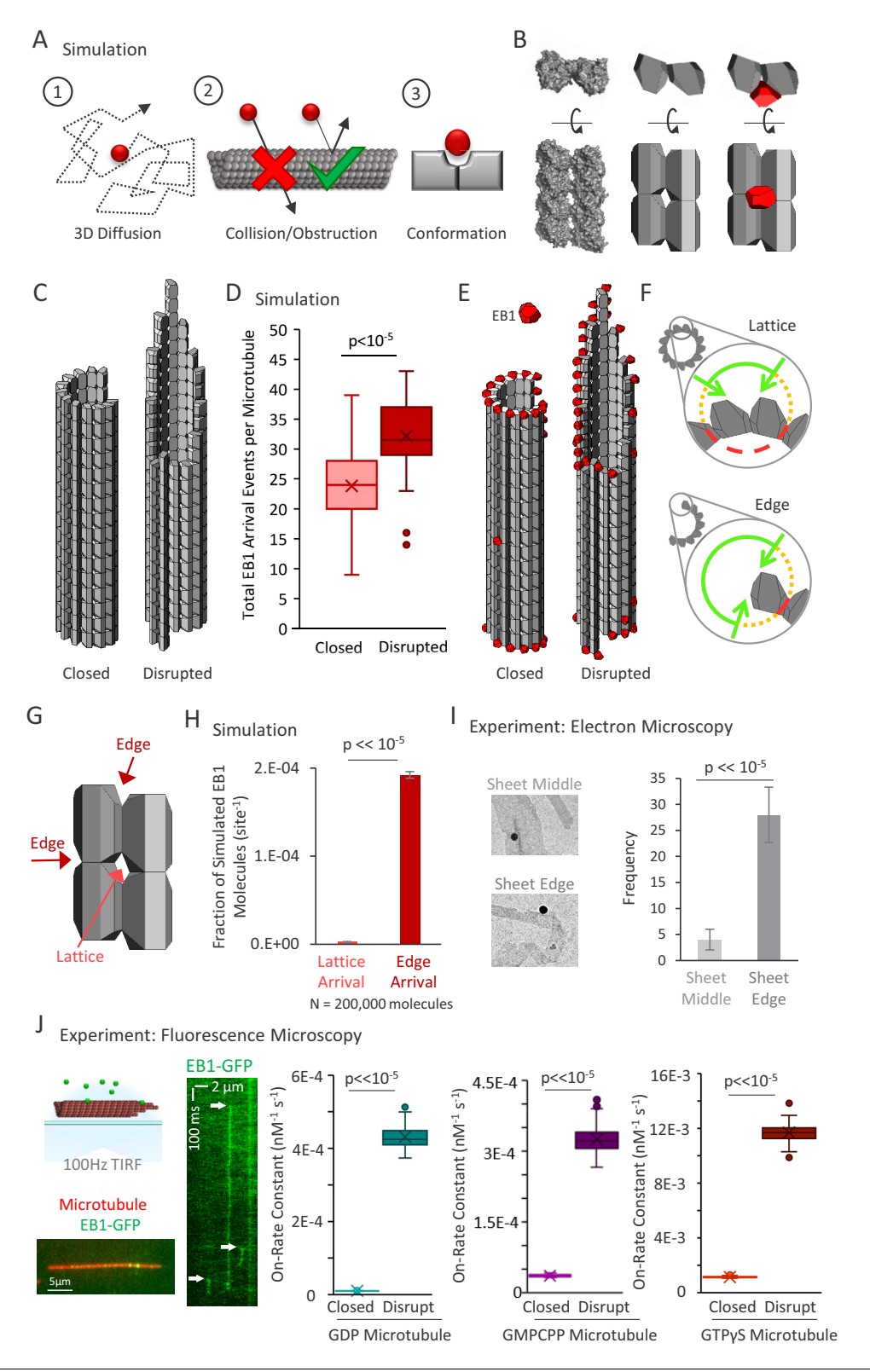

**Figure 3.** 3D diffusion-based simulations of EB1 predict an increased arrival rate at protofilament edge sites relative to closed, full lattice sites. (**A**) Simulation rules and parameters: (1) EB1 diffuses in three dimensions, (2) EB1 is obstructed by tubulin dimers and cannot pass through microtubules, (3) EB1 and tubulin dimer three-dimensional shapes reproduce EB1-tubulin binding interface. (**B**) Development of simulation shapes for EB1 and tubulin: Left: Cryo-EM reconstruction data (from PDB ID 3JAR) of tubulin dimers in a lattice: Left-top: Top-down view, with the lower portion being the outside

*Figure 3 continued*

of the microtubule. Left-bottom; Side view of four tubulin dimers, viewing the portion on the outside surface of the microtubule; Middle: Approximation of tubulin dimers for use in the simulation, derived from the cryo structure; Right: EB1 (red) also modeled as an approximation of cryo-structure data (not shown), with the binding interface correctly positioned at the pocket located between four adjacent tubulin dimers. (C) Two microtubule conditions used in the simulation analysis. Left: A closed, blunt-ended microtubule. Right: A 'disrupted-structure' microtubule with a tapered tip. Both the 'closed' and 'disrupted structure' configurations contain the same total number of tubulin dimers. (D) Results of 50 simulations of 4000 EB1 molecules each for each microtubule condition. Each data point in the box and whisker plot represents the total number of EB1 arrival events per microtubule for 4000 different simulated EB1 molecules. (E) Visualization of EB1 arrival events (red) on closed and 'disrupted-structure' microtubules from N = 10,000 simulated EB1 in each condition. (F) Illustration of the hypothesis that EB1's pocket-located binding site leads to a high local steric hindrance for EB1 binding to a lattice conformation (top), and a lower steric hindrance for EB1 binding to an edge conformation (bottom). Green lines: portion of the local volume with high accessibility to binding site. Yellow dotted lines: reduced accessibility volume. Red dashed lines: volume with no direct accessibility. In an edge conformation (bottom), the high accessibility region (green) is much larger than in the lattice conformation (top). (G) Dark Red arrows: Binding sites with two adjacent tubulin dimers, such as at the ends of protofilaments, or at exposed edges of protofilaments with no neighbor, are termed 'Edge' sites. Light red arrow: Sites with four adjacent tubulin dimers are termed 'Lattice' sites. Sites with only one tubulin dimer and with three adjacent tubulin dimers were not included in this analysis. (H) Fraction of simulated EB1 molecules that arrived at lattice sites (light red) or at edge sites (dark red). Data used was from the closed microtubule arrangement (panel C,E, left). Values were determined by dividing the total number of EB1 bound at any lattice or edge site by the total number of available lattice or edge sites, respectively. EB1 was ~70 fold more likely to bind a given edge site as compared to a given lattice site. (I) Left: example electron microscopy images of EB1 conjugated to gold beads on GDP microtubules. This image shows EB1-gold bound at a sheet edge, and one bound at a sheet middle (beads at ambiguous locations were conservatively classified as 'middle'). Right: Count of total number of sheet bound beads observed over all images, divided into 'Middle' of sheet and 'Edge' of sheet. (J) Far Left Top; Experimental setup, rhodamine-labeled microtubules are affixed to a coverslip (red), EB1-GFP is introduced in solution (green), and the sample imaged at 100 frames per second using total internal reflection fluorescence (TIRF) microscopy. Far Left Bottom; Sample image of EB1-GFP on the microtubule. Left: Kymograph of EB1-GFP with length along the x-axis and time down the y-axis. White arrows indicate EB1-GFP binding events, which appear as vertical streaks. Example shows atypically long EB1 association events, for clarity. The lower limit of the vertical streaks are the dissociation event of EB1-GFP from the microtubule. Right: EB1-GFP on-rate constant for closed and disrupted-structure microtubules in each nucleotide population.

DOI: https://doi.org/10.7554/eLife.48117.005

The following figure supplement is available for figure 3:

**Figure supplement 1.** Additional Data to Support *Figure 3*.

DOI: https://doi.org/10.7554/eLife.48117.006

To run the simulation, a microtubule was fixed in space, and then individual EB1 molecules were allowed to diffuse, starting from random positions far away from the microtubule (see Materials and methods). Two arrangements of microtubules, each with 207 tubulin dimers (average protofilament length ~16 dimers), were simulated to mimic 'closed' and 'disrupted-structure' pools of microtubules. For the closed microtubules, a microtubule with a blunt end was defined (*Figure 3C*, left), and for the 'disrupted-structure' model, a microtubule with a tapered end was defined (*Figure 3C*, right). Starting from a random position away from the microtubule (see Materials and methods), each EB1 molecule was then allowed to randomly diffuse either until it exceeded 2 μm away from the microtubule, or until the center of its binding interface was oriented properly and within 1 nm of each of its corresponding binding interface centers at any microtubule binding location (defined as an 'arrival event'). Once either of these conditions was achieved for a particular EB1 molecule, the simulation for that molecule was ended, and, for EB1 molecules with arrival events, their arrival position on the microtubule was recorded. We note that off-rates (or dwell times) were not determined as part of this simulation, as a binding event concluded the simulation for an individual microtubule.

First, we ran simulations to evaluate the arrival events of EB1 molecules onto the two microtubule configurations, regardless of EB1-tubulin binding configuration (e.g. 1–4 tubulin binding interfaces were allowed per EB1 molecule). We ran 50 simulations of 4000 EB1 molecules in each simulation for both the closed and the disrupted-structure microtubules, and calculated the total number of EB1 arrival events per microtubule in each simulation. We found that the mean number of successful arrival events of EB1 onto individual 'disrupted-structure' microtubules was ~33% higher than onto closed 'blunt' microtubules with the same number of tubulin subunits (*Figure 3D*; $p \ll 10^{-5}$, Binomial test). Strikingly, by examining the simulated arrival locations of individual EB1 molecules, we found that the EB1 molecules had arrived at nearly all of the sites at the microtubule end or along laterally exposed protofilaments, while very few arrived within a complete, closed lattice, both for the closed and the 'disrupted-structure' microtubules (*Figure 3E*, random representative subset of 10,000 simulated EB1 molecules).

Based on this observation, we hypothesized that the unique location of EB1's binding site in the 'pocket' between four tubulin dimers was responsible for the simulated 'structural recognition' of EB1 (*Figure 3F*, cutaway view – tubulin subunits above and below the section are not shown). Specifically, when a full lattice is present, the nature of the binding pocket between four tubulin dimers would restrict access of EB1 to its binding site, due to the high diffusional steric hindrance barrier generated by the nearby tubulin subunits, and the specific orientation that is required for EB1 to fit into this pocket and simultaneously bind at four locations (*Figure 3F*, top, short green arc shows restricted access). However, when there is a region of the microtubule with exposed protofilament edges, the steric hindrance barrier would be dramatically reduced, thus allowing for increased accessibility and ease of EB1 binding to the laterally exposed edges (*Figure 3F*, bottom, longer green arc shows increased access). This effect is likely even more dramatic than *Figure 3F* would suggest, since if this schematic were extrapolated to three dimensions, only a small wedge of a sphere would have high accessibility to an EB1 binding site for a closed lattice configuration. Importantly, a natural result of this model is that the on-rate of EB1 would be much higher at laterally (or longitudinally) exposed microtubule binding sites as compared to the binding pocket within a closed microtubule lattice.

To directly test the idea that the arrival rate of EB1 would be increased at laterally or longitudinally exposed microtubule binding sites, we ran simulations to compare the fraction of simulated EB1 arrival events onto closed-lattice microtubule-binding sites with four adjacent tubulin dimers (*Figure 3G*, light red arrow), to edge-located microtubule-binding sites, with only two adjacent tubulin dimers either horizontally or vertically (*Figure 3G*, dark red arrows). Thus, we specifically compared simulated EB1 arrivals to 2-tubulin vs 4-tubulin sites, excluding both 1-tubulin sites that could potentially have a very high off-rate, and 3-tubulin sites. This allowed us to limit our analysis to the partial binding EB1 sites that have been previously observed using electron microscopy (*Guesdon et al., 2016*), and that we observed using electron microscopy in our new work (*Figure 2F*).

Strikingly, in this simulation, the fraction of EB1 arrivals was ~70 fold higher for 2-tubulin sites with lower steric hindrance, such as those at protofilament edges (*Figure 3H*, dark red), as compared to 4-tubulin sites on the closed microtubule lattice (*Figure 3H*, light red; $p \ll 10^{-5}$, Chi-squared test). In order to compare these simulated arrival rates with the theoretical upper limit for our simulation, we modeled EB1 binding to a single, isolated tubulin heterodimer. In this instance, there would be no significant steric hindrance to binding, although arrival to the 1-tubulin site was still stereospecific, such that EB1 could not bind 'upside down'. We found that the fraction of EB1 arrivals to these 1-tubulin sites was 309-fold higher than the fraction of arrivals to full lattice 4-tubulin sites, and 4.6-fold higher than the fraction of arrivals to 2-tubulin edge binding sites. This difference clearly demonstrates the large impact of steric hindrance for the pocket-located binding site of EB1 in the simulation, especially when fully enclosed within the lattice.

## Electron microscopy experiments demonstrate binding of EB1-gold beads to microtubule sheet edges

To test this model experimentally, we further analyzed the binding of EB1-beads onto stabilized microtubule sheets from our electron microscopy experiments (*Figure 3I*, left). Qualitatively, we observed EB1-beads that bound to the stabilized microtubule sheets, both in the middle of the sheet, and on the edges of sheets (*Figure 3I*, left), as has been previously observed on the more transient microtubule sheets that are present at dynamic microtubule plus-ends (*Guesdon et al., 2016*). To quantify this observation, we counted the number of beads located at sheet edges as compared to the middle of sheets. Here, beads that were ambiguously located were conservatively classified as being at the middle of the sheet. We observed that, for sheet-like regions of microtubules, there were sevenfold more beads associated with sheet edges than with the middle of the sheet (*Figure 3I*, right; $p \ll 10^{-5}$, Chi-squared test). This suggests that EB1 can indeed bind at edge sites on a microtubule, despite the reduced number of inter-site binding partners. Further, the ratio of edge sites to total binding sites within a sheet is estimated at 2:14 (edge sites to total binding sites), and so by normalizing to the number of available sites in each case, this yields an observed ~49 fold preference for edge sites over lattice sites. This fold preference is on the order of the ~70 fold increase in EB1 on-rates for edge sites as compared to lattice sites from the 3D diffusion simulation (*Figure 3H*). We note that there was a lower reported ratio of edge to middle

position binding on dynamic microtubule sheets (1:18, *Guesdon et al., 2016*). However, for growing microtubules in the presence of free tubulin, the possibility that EB1 first binds to a sheet edge, and that this edge binding position is then converted to a sheet-middle position by subsequent addition of new tubulin subunits, cannot be ruled out. In contrast, for our stabilized microtubules with no free tubulin, binding in a sheet-middle position would require that EB1 binds directly to the sheet-middle position. Therefore, the mechanism for localization of EB1 to sheet-middle positions may be very different for stable microtubule experiments with no free tubulin, as compared to dynamic, growing microtubule experiments in the presence of free tubulin.

## Rapid frame-rate fluorescence microscopy experiments demonstrate increased on-rate of EB1-GFP to disrupted-structure microtubule pools

Our 3D single-molecule diffusion simulations predicted a ~ 70 fold rise in the fraction of EB1 arrivals to 2-tubulin edge structures relative to a 4-tubulin-pocket closed lattice conformation, due to a high diffusional steric hindrance barrier that frustrates EB1 from binding in the pocket-like interface between four adjacent tubulin dimers in the lattice. Thus, the simulated arrival rate of EB1 to its binding site on the microtubule was dictated by the structural state of the microtubule itself. To experimentally test this model prediction, we collected rapid frame-rate (100 Hz) movies using TIRF microscopy, and then tracked the association and dissociation events of individual EB1-GFP molecules with microtubules (*Figure 3J*, left; white arrows in EB1-GFP kymograph indicate EB1 association events). Because it was not possible to directly distinguish between EB1-GFP that was bound to a protofilament edge vs EB1-GFP that was bound at a closed lattice site in our TIRF experiments, we used our closed and disrupted-structure microtubule preparation protocols (*Figure 2*) to measure the difference in EB1-GFP association and dissociation rates between closed microtubules, and microtubules that likely had an increased number of exposed edge binding sites (disrupted-structure microtubules). The EB1-GFP on-rate was calculated in each case by fitting a decaying exponential to the dwell time histogram for individual binding events (*Figure 3—figure supplement 1A*), which allowed us to calculate the expected total number of binding events, including those with a very short dwell time (*Telley et al., 2009*) (See Materials and methods).

We found that the association rate for EB1-GFP was ~45 fold higher for disrupted-structure microtubules as compared to closed microtubules for the GDP microtubule pools (*Figure 3J*, center, teal; $p \ll 10^{-5}$; Z-test). This fold increase was similar to the normalized results for the electron microscopy EB1-bead studies (*Figure 3I*), suggesting that there was a substantial increase in edge-binding sites in the disrupted-structure pool as compared to the closed pool. For the GMPCPP microtubule pool, there was a ~ 12 fold increase in the EB1-GFP association rate constant in the disrupted-structure microtubule pool relative to the closed pool (*Figure 3J*, center, purple; $p < 10^{-64}$; Z-test), suggesting that the CaCl$_2$ treatments used to disrupt microtubule structure may be less efficient at producing edge-binding sites than the protocol for the GDP microtubule pools, which relied on damage initiated by Taxol treatment. Finally, the EB1-GFP association rate constant was ~3.5 fold higher in the disrupted-structure microtubule pool as compared to the closed microtubules for the GTPγS microtubules (*Figure 3J*, right, brown; $p = 1.7 \times 10^{-16}$; Z-test). This more moderate increase in the association rate constant may be due to a smaller difference in microtubule structure between the closed and disrupted-structure pools of GTPγS microtubules, as the closed microtubule pool itself was shown to exhibit structural disruption (*Reid et al., 2017*).

We then calculated EB1-GFP dwell times on the microtubule lattice from fits to our experimentally observed exponentially decaying dwell-time histograms (*Figure 3—figure supplement 1A*). We found that all pools of disrupted-structure microtubules had lower characteristic dwell-times than their respective closed pools (*Figure 3—figure supplement 1B*), suggesting that, while EB1 binding to protofilament edges may be rapid due to low steric hindrance, a twofold reduction in the number of EB1-tubulin bonds at the edge binding sites would lead to an increased EB1 off-rate. Consistent with this model, we found that EB1 dwell time distributions were best modeled as two exponential distributions (*Figure 3—figure supplement 1C,D*), one with a short dwell time (~20 ms), which could be associated with edge-bound EB1, and a second underlying distribution with a longer dwell time (~150 ms), which may correspond to lattice-bound EB1.

## In experiments and simulations, a tubulin face-binding protein does not exhibit microtubule structure recognition

Taken together, our data suggests that the pocket localization of EB1's binding site on the microtubule acts to limit access of EB1 to 4-tubulin lattice binding sites relative to 2-tubulin edge binding sites (*Figure 4A*, left vs right, lighter green arc shows large increase in accessibility). In contrast, this model predicts that a protein with a binding site on the face of a tubulin dimer would have similar binding site accessibility on both lattice and edge sites (*Figure 4B*, left vs right, lighter green arc shows small difference in accessibility). Thus, we predicted that a tubulin face binding protein would not show an increased number of arrival events to disrupted-structure microtubules.

To ask whether our simulation supported this prediction, we first performed tubulin 'face binding' simulations that had identical rules to the EB1 diffusion simulations (*Figure 4C*, 1–2), but with the exception that the protein binding site was located on the outer face of a tubulin dimer (*Figure 4C*, 3), rather than inside the four-tubulin pocket between dimers. Thus, we used our previous tubulin dimer and EB1 conformations, but instead moved the protein binding location to the outer face of the tubulin heterodimer, instead of within the four-dimer pocket between tubulin dimers (*Figures 4C*, 3, right). We then asked how the total number of face-binding molecule arrival events per microtubule in the closed and disrupted-structure microtubule simulations would be altered by changing this rule.

In previous EB1 simulations, the mean number of EB1 arrival events per microtubule was increased for the disrupted-structure microtubules relative to the closed microtubules (*Figure 4D*, red; $p=3.2\times10^{-28}$, Binomial test; 4000 EB1 molecules per simulation, 50 simulations). In contrast, we found that when the protein binding site was located on the tubulin heterodimer's outward face, the mean number of face-binding molecule arrival events for disrupted-structure microtubules was not substantially increased relative to the closed microtubules (*Figure 4D*, blue; $p=0.046$, Binomial test; 4000 face-binding molecules per simulation, 50 simulations), consistent with our hypothesis that the face-binding protein would not differentiate between edge and lattice binding sites. Similarly, the previous observation that simulated, pocket-binding EB1 molecules showed preferential arrivals to edge-proximal tubulin heterodimers (*Figure 4E*, red) was strikingly altered in the tubulin face-binding simulations: the face-binding protein arrival events were located non-specifically along the lattice, and with no enrichment on edge dimers (*Figure 4F*, blue; random representative subset of 10,000 simulated EB1 molecules). Thus, our simulations predicted that a tubulin face-binding protein would show similar rates of arrival to both closed and disrupted-structure microtubule populations, due to a similar steric hindrance to binding in both cases.

To experimentally test this prediction, we used a monomeric Kif5B-GFP kinesin construct in the presence of 10 mM AMPPNP, to allow for microtubule binding analysis of the monomeric kinesin (K339-GFP) (*Case et al., 2000*; *Tomishige and Vale, 2000*). The Kinesin-1 Kif5B protein has been shown to dock with a stoichiometry of one kinesin motor per tubulin heterodimer, and binds to the outer surface of the α and β tubulin monomers, rather than the lateral surfaces, in electron microscopy experiments (*Hirose and Amos, 1999*; *Kikkawa et al., 2000*; *Kozielski et al., 1998*; *Löwe et al., 2001*; *Marx et al., 2006*; *Moores et al., 2002*; *Nogales et al., 1999*; *Rice et al., 1999*; *Sosa et al., 1997*). Thus, we used the Kif5B-GFP construct to determine whether a tubulin face-binding protein would show differential binding to closed and disrupted-structure microtubule populations, as was observed for the EB1-GFP protein (*Figure 2B*).

We first used 10 nM Kif5B-GFP, and compared overall binding of the Kif5B-GFP to closed and disrupted-structure pools of GDP microtubules, similar to the procedure for our EB1-GFP experiments (*Figure 2B*). Qualitatively, and in contrast to the EB1-GFP experiments, the degree of binding between the closed and disrupted-structure pools of GDP microtubules appeared similar (*Figure 4G*, left). Quantification of the TIRF images demonstrated that the average Kif5B-GFP/tubulin intensity ratio for the disrupted-structure pool of GDP microtubules was not higher than in the closed microtubules ($p=1$; one-sided t-test), but rather appeared to be slightly lower than in the closed microtubules ($p=2.7\times10^{-7}$, 2-sided t-test), perhaps suggesting a slight suppression of Kif5B-GFP binding to microtubules with disrupted-structure structures. Similar results were observed using a lower concentration of Kif5B-GFP (2.5 nM, *Figure 4G*, right; $p<10^{-15}$, two-sided t-test). These results suggest that the binding location of EB1 on microtubules may dictate its preferential binding

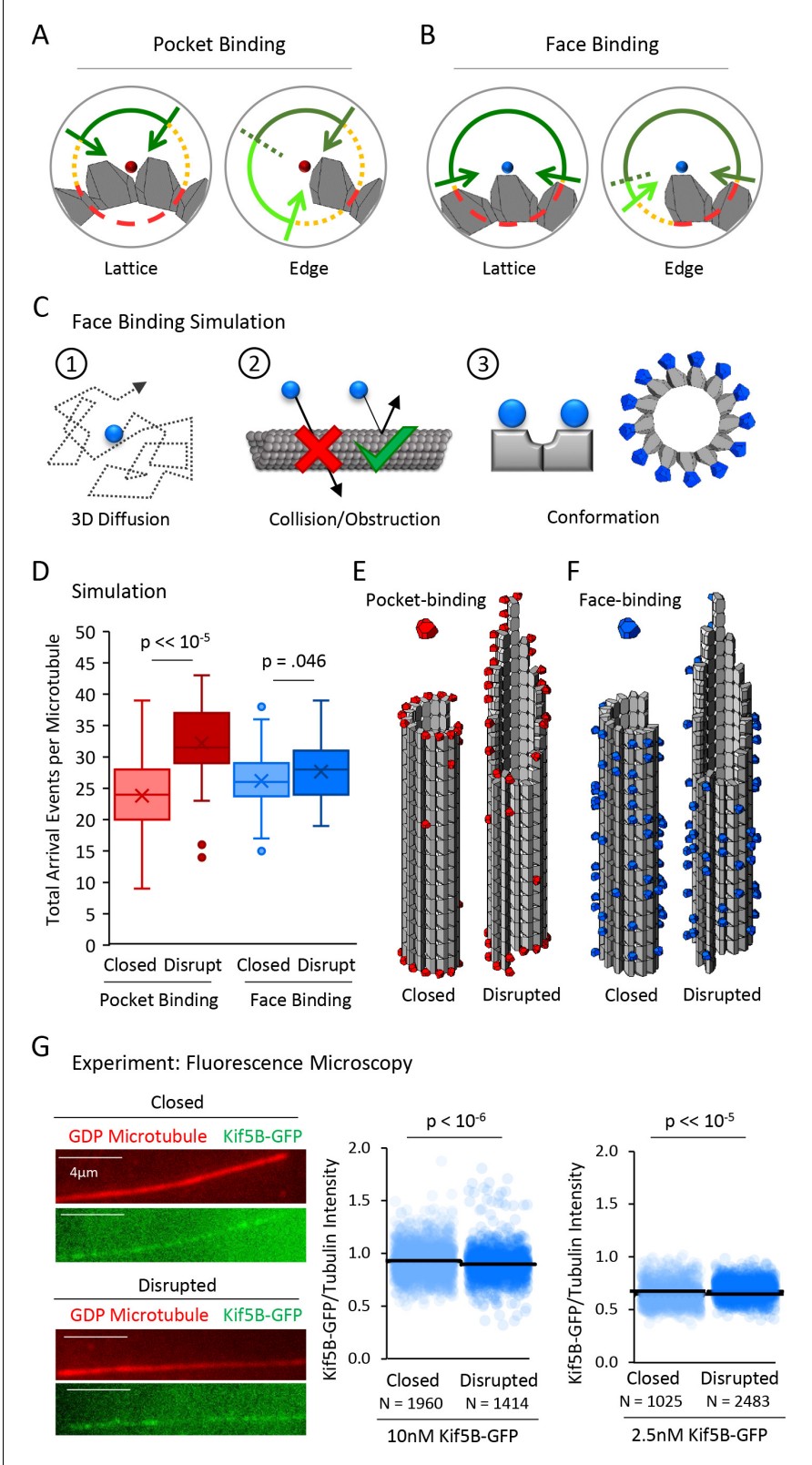

**Figure 4.** A tubulin face binding protein does not demonstrate microtubule structure recognition in experiment and simulation. (**A**) Hypothesis schematic showing large increase in direct access to a pocket binding site (dark green + light green) upon change from lattice conformation (left) to an edge conformation (right) for a pocket-binding protein. (**B**) Face binding hypothesis model predicts only a small change in direct access to a face binding

*Figure 4 continued on next page*

*Figure 4 continued*

site upon change from lattice conformation (left) to an edge conformation (right). (C) Simulation rules and parameters for face binding are identical to a pocket binding simulation with respect to (1) 3D diffusion, and (2) obstruction of the binding molecule by tubulin molecules. (3) The only change in the simulation was in the location of the binding site (left), with the arrangement of simulated face-binding protein on a microtubule at right. (D) Total arrival events per microtubule for a simulated pocket binding protein (EB1, red) and face binding protein (blue), showing molecular arrival events on closed (light red/light blue) and 'disrupted-structure' (dark red/dark blue) microtubule conformations. Results of 50 simulations of 4,000 EB1 molecules each for each microtubule condition. Each data point in the box and whisker plots represents the total number of EB1 arrival events per microtubule for 4000 different simulated EB1 molecules. (E) Representative visualizations of arrival event location distribution of pocket binding protein such as EB1 (red) on closed and disrupted-structure microtubules. (F) Representative visualizations of arrival event location distribution of face binding protein (blue) on closed and disrupted-structure microtubules, illustrating the loss of preference for edge-located binding sites. (G) Left: Representative images of Kif5B-GFP (green), which is a tubulin face binding protein, on GDP microtubules (red) (scale bar 4 μm). Right: Cloud plots of Kif5B-GFP/Tubulin intensity ratio for individual microtubules. Each semi-transparent circle is the data point from a single microtubule. Solid lines are the average Kif5B-GFP/Tubulin intensity ratio for each population.

DOI: https://doi.org/10.7554/eLife.48117.007

to the disrupted-structure pool of microtubules relative to the closed microtubules, and that binding to the face of a tubulin dimer by Kif5B does not yield this preferential binding behavior.

## EB1 tip tracking by microtubule structure recognition

We then asked how the rapid binding of EB1 onto protofilament edges could be integrated with current models for tip tracking behavior on growing microtubule ends. It has been previously shown that EB1-GFP is concentrated in a location on growing microtubule ends that is slightly behind the distal tip (*Maurer et al., 2014*). This EB1-GFP fluorescence distribution has been explained via a model in which EB1 first binds the microtubule with high affinity in a location that is penultimate to the microtubule tip, and then, over time, a conformational change occurs on the microtubule that lowers the affinity of EB1 for this binding site (*Maurer et al., 2014*). Because EB1 binds the microtubule with high affinity in a location that is penultimate to the microtubule tip, one hypothesis is that EB1 binds with high affinity to a GDP-$P_i$ lattice, which would be localized behind the tip of the microtubule due to a delay in hydrolysis of newly added GTP-tubulin subunits at the microtubule tip (*Zhang et al., 2015*). In this model, the conversion of the microtubule lattice from a GDP-$P_i$ state to a GDP state would then lower the affinity of EB1 for its binding site. Thus, we analyzed the overall binding of EB1-GFP to GDP-$P_i$ microtubules as compared to GDP microtubules, to determine whether EB1-GFP binds with high affinity to GDP-$P_i$ microtubules, and with a lower affinity to GDP microtubules.

It has been previously shown that the $K_d$ of $P_i$ for GDP is ~25 mM (*Carlier et al., 1988*; *Raw et al., 1997*). Furthermore, the depolymerization rate of GDP microtubules is substantially slowed in the presence of 50 mM of inorganic phosphate, indicating that $P_i$ binds to GDP-tubulin under these conditions (*Carlier et al., 1988*). Therefore, under these conditions, we compared the binding of EB1-GFP to GDP and GDP+$P_i$ microtubules to determine whether EB1 exhibits preferential binding to GDP+$P_i$ microtubules (*Figure 5A*; see Materials and methods). Surprisingly, EB1-GFP binding appeared to be reduced on GDP+$P_i$ microtubules relative to GDP microtubules, for both closed and disrupted-structure microtubules (*Figure 5B*). By quantifying EB1-GFP binding on each microtubule type using our previously described automated analysis tool (*Reid et al., 2017*), we found that there was a −53% reduction in binding of EB1-GFP to closed GDP+$P_i$ microtubules relative to closed GDP microtubules (p<<$10^{-5}$, t-test) (*Figure 5C*, experimental repeats *Figure 5—figure supplement 1A*).

In the KPO$_4$ experiments (*Figure 5A–C*), while the molarity of the salts in the experiments were matched (55 mM KPO$_4$ vs 55 mM KCl), the ionic strengths of the reaction mixtures were mismatched, which could influence EB1 binding to the lattice. Therefore, we performed further experiments using an analog of phosphate, AlF$_4^-$ (*Antonny and Chabre, 1992*; *Carlier et al., 1988*; *Petsko, 2000*). Here, the molarity and ionic strengths of the salts included in both experiments were

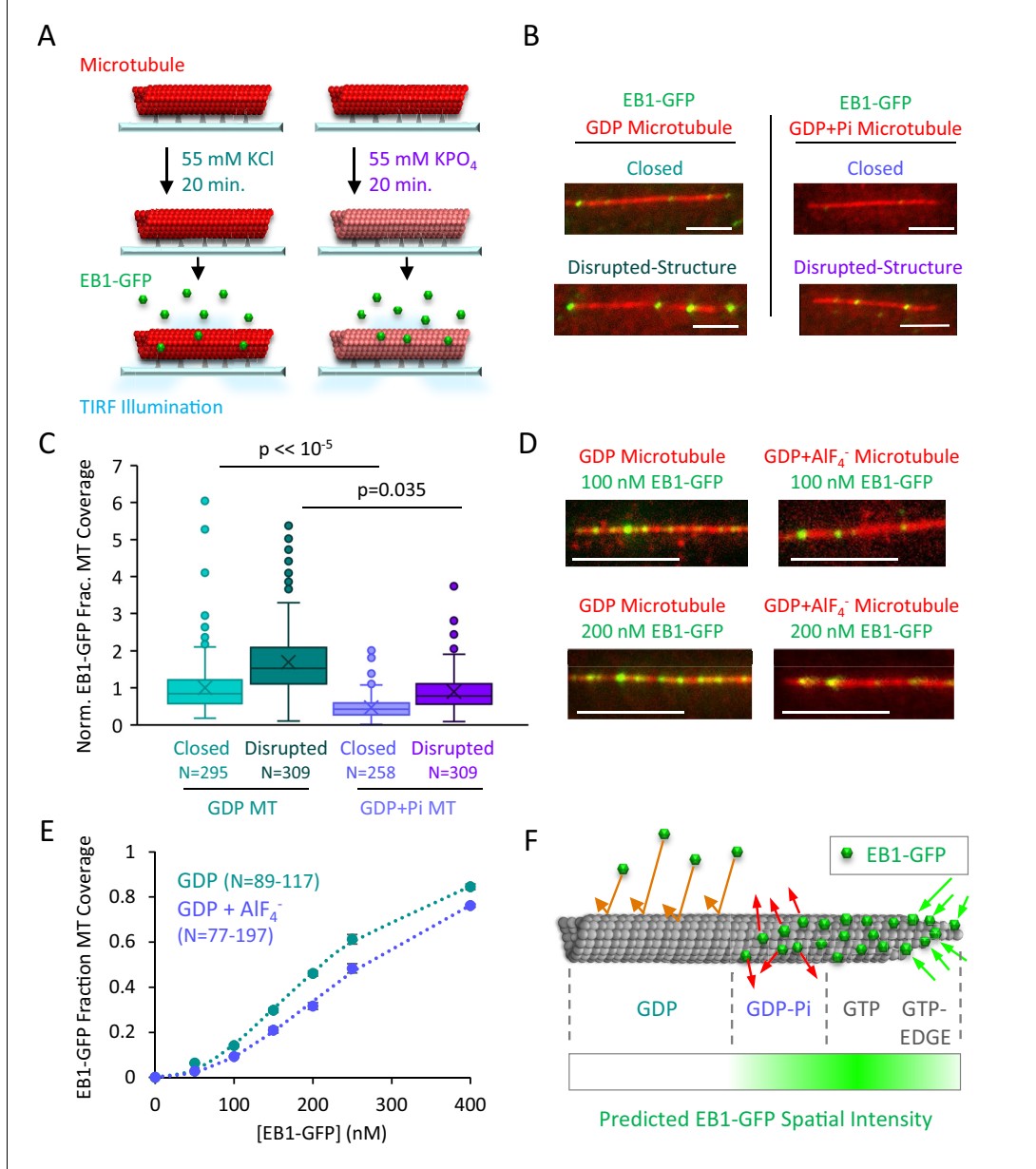

**Figure 5.** EB1-GFP binding is suppressed on GDP-$P_i$ microtubules. (**A**) Experiment to compare EB1-GFP binding to GDP and GDP+$P_i$ microtubules (see Materials and methods). (**B**) Representative images of EB1-GFP binding to GDP and GDP+Pi microtubules (scale bar 5 μm). (**C**) Quantitative binding comparison, based on EB1-GFP fractional coverage area of microtubules. All values normalized to the average fractional coverage area in the closed GDP microtubule population. Sample sizes represent number of independent images analyzed. (**D**) Representative images of EB1-GDP binding to closed microtubules in the presence and absence of the phosphate analog AlF$_4^-$ (*Antonny and Chabre, 1992*; *Carlier et al., 1988*; *Petsko, 2000*) (scale bar 5 μm). (**E**) Quantitative analysis of EB1-GFP binding under conditions of increasing EB1 concentrations. As previously described, the EB1 binding data was best fit using a cooperative binding curve (*Lopez and Valentine, 2016*; *Zhu et al., 2009*), which demonstrated a ~ 2 fold lower affinity of EB1 for the GDP-Pi-Analog relative to the GDP microtubules. (**F**) Implementation of structural recognition into a two-state model for EB1 binding and microtubule tip structure maturation.

DOI: https://doi.org/10.7554/eLife.48117.008

The following figure supplement is available for figure 5:

**Figure supplement 1.** Additional Data to Support *Figure 5*.

DOI: https://doi.org/10.7554/eLife.48117.009

identical (52.5 mM KCl and 2 mM NaF for both experiments). However, 10 μM AlCl$_3$ was included in the phosphate analog experiments, allowing the Al$^{3+}$ and F$^-$ ions form the Pi analog AlF$_4^-$, which mimics a phosphate group, but with 1000-fold higher affinity (*Antonny and Chabre, 1992*; *Carlier et al., 1988*; *Petsko, 2000*).

Similar to the KPO$_4$ experiments, we observed reduced EB1 binding in the GDP+AlF$_4^-$ experiments as compared to the GDP-alone experiments (*Figure 5D*). We then quantified EB1-GFP binding using our previously described automated analysis tool (*Reid et al., 2017*) under conditions of increasing EB1 concentrations (*Figure 5E*). As previously described, the EB1 binding data was best fit using a cooperative binding curve (*Lopez and Valentine, 2016*; *Zhu et al., 2009*), and so we fit the results in each case to a Hill equation, which demonstrated a ~ 2 fold lower affinity of EB1 for the GDP-Pi-Analog relative to the GDP microtubules (*Figure 5E*, see Materials and methods; $K_{d, GDP}$=1.9x10$^5$ nM, $n_{GDP}$ = 2.2; $K_{d,Analogue}$=4.2x10$^5$ nM, $n_{Analogue}$ = 2.3).

These results could perhaps be explained by previous cryo-electron microscopy findings in which it was demonstrated that the GDP-P$_i$ microtubule structural state is distinct from that of GDP microtubules (*Manka and Moores, 2018b*). Further, published cryo-EM structures of microtubules coassembled with EB3 show that EB1 may promote a compacted microtubule lattice with a lattice twist, once it is bound to the microtubule (*Zhang et al., 2015*). This structural transition could account for the ultimate destruction of the stable 4-tubulin binding site of EB1 as the lattice-bound tubulin subunits transition from GTP to GDP-Pi, and ultimately to a GDP conformations, perhaps promoted by the binding of EB1 to the microtubule. Thus, we conclude that EB1 does not likely directly bind to GDP-P$_i$ microtubules, but, instead, that hydrolysis of GTP-tubulin to GDP-P$_i$ -tubulin could initiate formation of a low-affinity EB1 binding site.

Our finding that EB1 binds with low affinity to GDP-P$_i$ lattices is perhaps unexpected based on previous dynamic microtubule experiments in which it was observed that Mal3-GFP was bound to the entire dynamic microtubule lattice in the presence of GTP together with BeF$_3^-$ (*Maurer et al., 2011*). Here, while BeF$_3^-$ is thought to replace the dissociated phosphate on the GDP lattice, perhaps mimicking a GDP-P$_i$ state, it may be that BeF$_3^-$ instead acts to lock the dynamic microtubule lattice into a GTP-like state in the presence of excess GTP, which is required in a dynamic microtubule assay. However, GTP was absent in our GDP-P$_i$ and GDP-Pi-Analog microtubule experiments, thus ensuring that phosphate was bound only to the GDP lattice.

Thus, our GDP+P$_i$ data, combined with our protofilament edge binding data, led us to hypothesize that the process of EB1 binding at the growing microtubule end could be characterized by (1) rapid EB1 binding to edge sites at the distal end of the microtubule, followed by maturation into a tight EB1 binding configuration through the addition of new GTP-tubulin subunits at locations of edge-bound EB1, which would stabilize EB1 binding by completing its 4-tubulin pocket binding configuration (*Figure 5D*, right), and (2) subsequent destruction of the stable 4-GTP-tubulin pocket binding site by hydrolysis of GTP-tubulin into GDP-P$_i$ (*Figure 5D*, center), and, ultimately, GDP-tubulin, which may be accelerated by the binding of EB1 into its 4-tubulin pocket binding configuration (*Maurer et al., 2014*; *Zhang et al., 2015*; *Zhang et al., 2018*).

We predict that rapid binding to protofilament edge sites at tapered microtubule ends would remain consistent with observations that the highest density of EB1-GFP is 80–160 nm behind the most distal end of the microtubule (*Maurer et al., 2014*). Here, although new edge sites are continuously generated as a result of tubulin subunit additions to growing microtubule plus-ends, the density of edge sites on an open, tapered microtubule sheet is, by definition, seven-fold lower than that of lattice sites on a closed microtubule tube (ratio of edge-binding-sites to total-binding-sites per tubulin layer for an open microtubule sheet is 2:14). Thus, even with rapid loading of EB1 onto edge sites at the most distal positions on the microtubule tip, the peak in EB1-GFP intensity for a plus-end EB1-GFP comet would be penultimate to the tapered tip, in the location where a high density of EB1 molecules are bound to the much higher density, four-GTP-tubulin pocket lattice binding sites, perhaps via addition of new, incoming GTP-tubulin subunits onto the location of previously edge-bound EB1 molecules. Consistent with this prediction, by generating simulated microscope images of microtubule tips with complete occupancy of EB1-GFP at distal tip edge sites and an exponentially decaying occupancy of EB1-GFP away from the distal tip on the closed microtubule lattice sites (*Figure 5—figure supplement 1B*), we observed that the highest density of EB1-GFP lagged behind the distal microtubule end (*Figure 5—figure supplement 1C*), as previously described (*Maurer et al., 2014*). Thus, in a model where EB1 is preferentially targeted to edge binding sites at

tapered microtubule ends, the reduced density of edge binding sites relative to closed microtubule lattice binding sites could explain the penultimate location of EB1 at growing microtubule ends (*Figure 5D*).

## Blunting of tapered microtubule tip structures correlates with suppressed EB1 binding in cells

An important prediction of a structural-recognition model, in which EB1 tip tracking is facilitated by tapered microtubule ends (*Figure 5D*), is that the pruning of extended, tapered tip structures at growing microtubule ends, leading to blunt ends with fewer protofilament-edge EB1-binding sites, would suppress EB1 tip tracking in cells. To test this idea, we treated LLC-Pk1 cells that expressed Tubulin-GFP (*Rusan et al., 2001*) with increasing, low-dose concentrations of the microtubule desta-bilizing drug Vinblastine, to determine whether we could alter the structure of the growing microtubule plus-ends. While Vinblastine does not alter the hydrolysis rate of GTP-tubulin (*Castle et al., 2017*), it interferes with tubulin assembly into protofilaments (*Gigant et al., 2005*), and could thus suppress the tapered multi-protofilament extensions that have been observed at rapidly growing microtubule plus-ends (*Chrétien et al., 1995*; *Coombes et al., 2013*; *Guesdon et al., 2016*).

We treated LLC-Pk1 cells with low concentrations of Vinblastine, which still allowed for robust growth of microtubules (*Figure 6A*). Then, we collected line scans of the drop-off in tubulin-GFP fluorescence at growing microtubule plus-ends, to assess whether microtubule tip structures were altered (*Figure 6B*). Qualitatively, it appeared that tubulin-GFP fluorescence at the tips of microtubules in the control cells dropped off more slowly to background as compared to the 10 nM Vinblastine-treated cells, in which the fluorescence dropped off more abruptly (*Figure 6B*, blue vs yellow). To quantify this observation, we fit a Gaussian error function to the fluorescence intensity drop-off at each microtubule end, as previously described (*Coombes et al., 2013*; *Demchouk et al., 2011*), which allowed us to estimate tip tapering that arises as a result of protofilament length variability at the microtubule ends. Consistent with our qualitative observations, we found that the microtubule tip standard deviation was significantly reduced with Vinblastine treatments as low as 1 nM (*Figure 6C*; Control vs 1 nM: p=0.004, Z = 2.89; Control vs 10 nM: $p=6\times10^{-5}$, Z = 4.01), indicating that treatment with the microtubule destabilizing drug Vinblastine acted to prune protofilament extensions at growing microtubule plus-ends, eliminating tapered tips with extensive protofilament-edge binding sites. Indeed, tip standard deviations in 10 nM Vinblastine approached the point spread function of the microscope, similar to the blunt ends of GMPCPP microtubules (*Figure 6C* (yellow) vs *Figure 1D*), suggesting that extended protofilaments and associated tip tapering at growing microtubule ends had been eliminated at higher Vinblastine concentrations.

We then asked whether blunting of the microtubule tip structure, leading to a reduced number of protofilament-edge binding sites, would be correlated with the efficiency of EB1 tip tracking. Thus, we used an LLC-Pk1 strain with EB1-GFP (*Piehl and Cassimeris, 2003*), and measured the peak intensity of EB1-GFP at growing microtubule ends, relative to lattice bound EB1-GFP (*Figure 6D*; see Materials and methods). Consistent with qualitative observations (*Figure 6D*, bottom, white arrows), we found that peak EB1-GFP intensity at microtubule ends was reduced with increasing Vinblastine concentrations (*Figure 6E*, $p<<10^{-5}$, t-test), suggesting that, with flattened end structures, EB1-GFP could not efficiently target to growing microtubule plus-ends. To ensure that the reduced EB1 targeting was not a consequence of shorter EB1-GFP comet lengths in Vinblastine treatment, we then compared the peak intensity of EB1-GFP comets at similar EB1-GFP comet lengths between controls and in 1 nM Vinblastine (Control comet length: 0.60 ± 0.06 μm, n = 62; 1 nM Vinblastine comet length: 0.59 ± 0.07 μm, n = 90; (mean ± SD)). Here, while the EB1-GFP comet lengths were statistically indistinguishable in our samples (p=0.89, t-test), we nevertheless observed a significant decrease in peak comet intensity in the Vinblastine-treated cells relative to the same-comet-length control cells ($p=4.5\times10^{-6}$, t-test, *Figure 6—figure supplement 1A*).

Finally, to verify that the reduced peak EB1-GFP comet intensity in Vinblastine was not the result of an overall lower affinity of EB1 for microtubules in the presence of Vinblastine, we measured EB1-GFP binding to closed in vitro GDP microtubules in the presence of increasing Vinblastine concentrations (*Figure 6F*). Importantly, and in contrast to the reduced peak intensity of EB1-GFP comets on dynamic microtubules in cells, the binding of EB1-GFP to closed GDP microtubules was similar regardless of Vinblastine concentration (*Figure 6G*, no significant differences, t-test; experiment repeats, *Figure 6—figure supplement 1B*). We conclude that the efficiency of cellular EB1-GFP tip

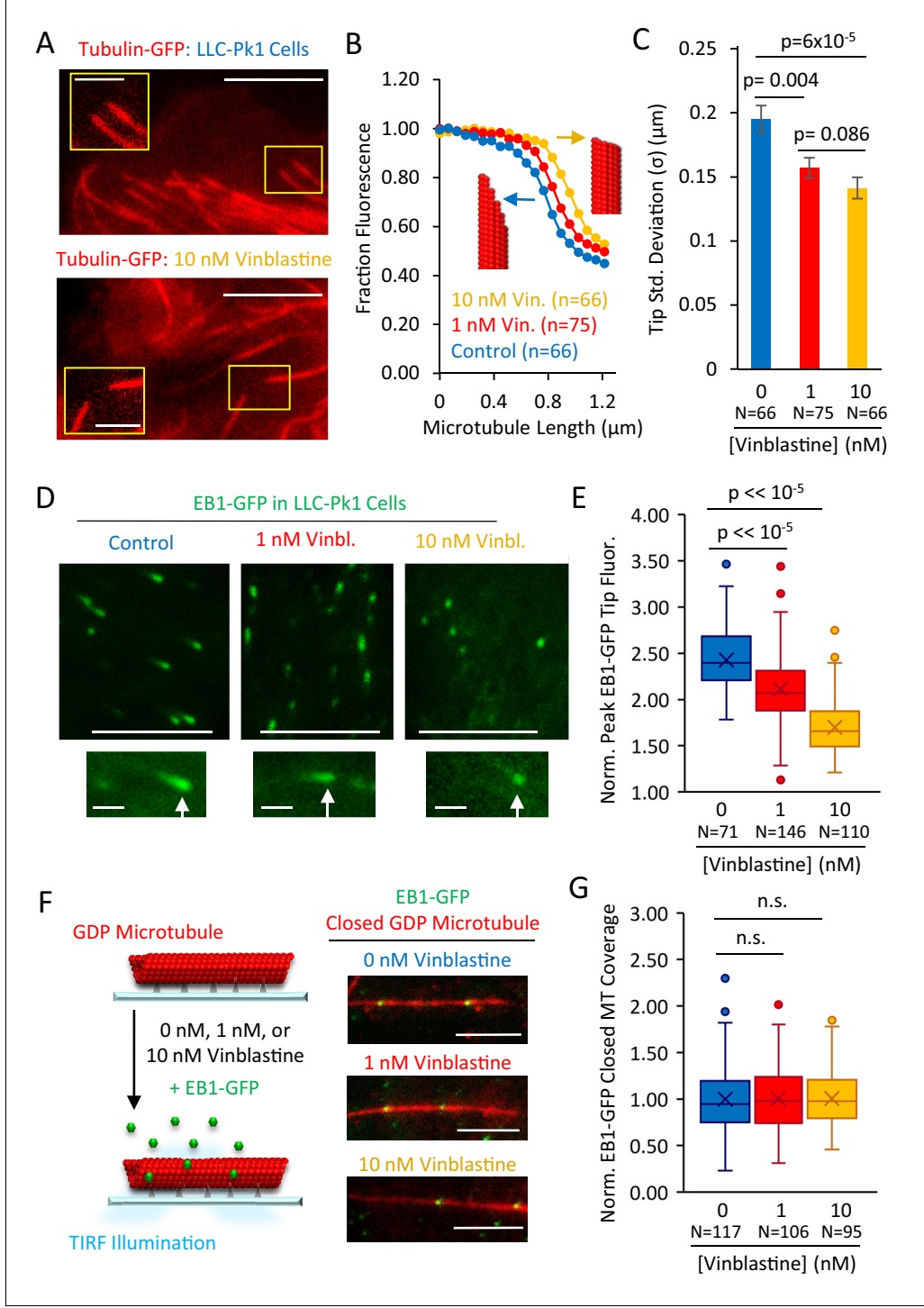

**Figure 6.** Structural recognition facilitates EB1 tip tracking at growing microtubule plus-ends in cells. (**A**) Representative images of Tubulin-GFP expressed in LLC-Pk1 cells with inset detail of growing microtubule tips (control cells, top, 10 nM Vinblastine treated, bottom; see Materials and methods) (scale bar 5 µm, insets 2 µm). (**B**) Average linescans that show the dropoff in intensity of tubulin at the tips of microtubules in control and Vinblastine treated cells. A sharper drop-off (yellow) is suggestive of a blunt microtubule end, and a slower dropoff (blue), is suggestive of a more tapered microtubule end, as shown by the red microtubule cartoons. (**C**) Tip standard deviations as estimated by fitting to the graphs in panel F. (**D**) Representative images of EB1-GFP expressed in LLC-Pk1 cells (top) with inset EB1-GFP comets at growing microtubule tips (bottom) (control cells, left; 1 nM Vinblastine treated, center; 10 nM Vinblastine treated, right; see Materials and methods) (scale bar 5 µm,

*Figure 6 continued on next page*

*Figure 6 continued*

comet detail scale bar 1 μm). (**E**) Peak EB1-GFP fluorescence normalized to lattice EB1-GFP fluorescence. (**F**) Experiment (left) and representative images (right) of in-vitro experiment to quantify binding of EB1-GFP to closed GDP microtubules in the presence of increasing concentrations of Vinblastine (scale bar, 5 μm). (**G**) Quantitative binding of EB1-GFP to closed GDP microtubules, based on EB1-GFP binding area on microtubules. All values normalized to the average binding area in the 0 nM Vinblastine population. Sample sizes represent number of independent images analyzed (p=0.91, t-test 0 nM vs 1 nM Vin.; p=0.88, t-test 0 nM vs 10 nM Vin.).
DOI: https://doi.org/10.7554/eLife.48117.010
The following figure supplement is available for figure 6:

**Figure supplement 1.** Supplemental Data to support *Figure 6*.
DOI: https://doi.org/10.7554/eLife.48117.011

tracking is disrupted with Vinblastine treatment, perhaps due to the blunting of tapered tip structures at growing microtubule ends, which cuts the availability of high-accessibility EB1 protofilament-edge binding sites. However, we cannot exclude that other structural or chemical cellular responses to Vinblastine may be at play as well.

## Discussion

Taken together, our results suggest a new model to explain EB1 tip tracking at growing microtubule plus-ends, in which EB1 recognizes and rapidly binds to the protofilament edges at open, tapered microtubule end structures. The subsequent addition of new GTP-tubulin subunits may then facilitate the maturation of EB1 into a high-affinity binding state on the closed GTP-tubulin lattice. Ultimately, tubulin hydrolysis into a GDP-Pi or GDP state destroys the EB1 binding site, catalyzing its release from the microtubule (*Figure 5F*).

We found that EB1 dwell time distributions on our disrupted structure microtubules were best modeled as two exponential distributions (Fig S2C,D), including one with a short dwell time (~20 ms), which could be associated with edge-bound EB1. In evaluating this dwell time relative to the kinetics of αβ-tubulin association, in 10 μM tubulin it has been estimated that a free tubulin subunit would arrive to the growing end of the microtubule as rapidly as every ~2 ms (*Gardner et al., 2011*), or as slowly as every ~10 ms (*Mickolajczyk et al., 2019*). Therefore, even the relatively short dwell times of edge-bounded EB1 molecules would be long enough to allow for multiple tubulin subunit arrivals while EB1 is bound to a protofilament edge. These kinetics suggest that if EB1 first binds to a protofilament edge, subsequent tubulin subunit arrivals could lock EB1 into its stable 4-tubulin pocket binding site. This pathway to establishing EB1 binding into its 4-tubulin pocket binding location would be efficient both for EB1, in which the steric hindrance for its direct binding into a 4-tubulin pocket was nearly insurmountable in our diffusion simulations, and for incoming tubulin subunits, which would be stabilized longitudinally and/or laterally by binding to EB1 in addition to other protofilament-bound GTP-tubulin subunits.

Overall, we propose that structural recognition by EB1 could promote rapid loading of EB1 at the open, tapered growing microtubule plus-end with its more easily accessible edge sites. Specifically, our results predict that the on-rate for EB1 binding to the closed microtubule lattice would be extremely low, due to very high steric hindrance and low accessibility. In contrast, we predict that the EB1 on-rate would be ~70 fold (~6000%) higher at tapered, growing microtubule plus-ends, due to the presence of easily accessible, low steric hindrance sites at exposed protofilament edges. This was the most surprising conclusion from our diffusion simulations: while we expected that tubulin edges would be more accessible to EB1 binding, the enormous magnitude of this preference was surprising to us. Importantly, the binding site geometries and diffusion coefficients used in the simulation were constrained by prior electron microscopy reconstructions and molecule size, and so the molecular diffusion simulations did not rely on any free parameters.

After binding to an edge-site, EB1 would likely undergo a higher off-rate as a result of the partial binding interface that is inherent to these edge-sites. However, our results show an increase in steady-state binding to disrupted microtubule structures, regardless of nucleotide state (*Figure 2C–E*), suggesting that the overall affinity of EB1 may be higher on disrupted structures, regardless of the likely increase in off-rates. This increase in affinity may be due to a stronger effect of disrupted

microtubule structure in accelerating the EB1 on-rate, as compared to increasing the EB1 off-rates. For example, we observed a ~ 45 fold increase in EB1 on-rates to disrupted GDP microtubule structures relative to closed microtubule structures (*Figure 3J*), while the 'short' EB1 dwell times, which could be associated with edge-bound EB1 on GDP microtubules, were only ~8 fold faster than the lattice-bound 'long' dwell times (*Figure 3—figure supplement 1D*), resulting in a net affinity increase of ~5 fold.

The effect of microtubule structure on microtubule-associated protein binding preference is not likely restricted to EB1, as we find that this preference stems from the physical accessibility of microtubule binding sites, and thus may be pertinent to any polymer binding protein. We expect that the magnitude of this effect will be strongly impacted by the location of the protein binding site on the microtubule lattice, as suggested by the difference in our pocket vs face binding simulations. Here, any protein that has its binding site located in the dip between protofilaments or at the crease between stacked subunits could potentially display microtubule structural recognition. For example, the microtubule-associated protein Doublecortin (DCX) has a similar binding site to EB1. DCX has been reported to recognize microtubule curvature, a configuration that would require an incomplete lattice with exposed protofilament edges to allow for outward protofilament curvature (*Bechstedt and Brouhard, 2012*; *Bechstedt et al., 2014*; *Fourniol et al., 2010*). Similarly, microtubules grown in the presence of CLIP-170 have been associated with the growth of curved oligomers (*Arnal et al., 2004*), and it has been shown that microtubule lattice defects may be recognized by CLIP-170 to stimulate microtubule rescue (*de Forges et al., 2016*). Thus, recognition of microtubule curvature or lattice defects by microtubule-associated proteins could rely, at least in part, on an increased arrival rate to exposed protofilament edge sites. Further, a recent electron microscopy study reported that the tips of growing microtubules within mitotic spindles display a gentle outward curvature of individual protofilaments (*McIntosh et al., 2018*). This outward curvature could enhance a structural recognition mechanism by increasing the availability of exposed protofilament edge sites at growing microtubule ends, due to the increased separation between individual protofilaments. In contrast, proteins such as TPX2, which binds in part across the faces of α and β tubulin subunits (*Zhang et al., 2017*), and TOG1-domain containing proteins such as XMAP215 and Stu2, where the TOG1 domain binds predominantly on the β-tubulin face (*Byrnes and Slep, 2017*), may be less likely to display structural recognition.

As noted above, because outward protofilament curvature at growing microtubule ends is necessarily associated with exposed protofilament edges, we cannot exclude the possibility that EB1 may also preferentially bind to curved protofilaments (*Guesdon et al., 2016*). However, we find that decreased steric hindrance can increase the arrival rate of EB1 to protofilament edges by ~70 fold (~6000%, *Figure 3*), while no similar effect would apply to protofilament curvature. Further, we did not routinely observe outward protofilament curvature at holes and defects within the lattice (*Figure 2B*), suggesting that the increased affinity of EB1 to disrupted lattice structures (*Figure 2C*) was not a result of outwardly curved protofilaments that were present within the microtubule lattice.

Our simulations model only one microtubule binding domain of EB1, yet EB1 is typically a homodimer. We anticipate that EB1's dimerization activity would not alter the observed microtubule structural preference in our simulations, but would likely increase the dwell time of edge-associated EB1, as once one domain binds to the more easily accessible protofilament edge sites, EB1's second binding domain would be constrained to a small area, and therefore much more likely to bind, even within a 4-tubulin pocket, due to the high local availability of binding sites. Thus, the EB1 homodimer as a whole would become more stably bound to the microtubule. EB1 dwell time was not included in our current simulations, as the simulation for a particular molecule was ended upon arrival to the microtubule lattice, but future iterations of our simulation could model dwell time to test this idea.

In conclusion, we find that a high steric hindrance barrier impedes EB1 from binding directly into the pocket-like interface between four adjacent tubulin dimers in the lattice, and that this barrier is markedly reduced at tapered microtubule ends. Protofilament edge-binding greatly increases the arrival rate of EB1 to growing microtubule plus-ends, which facilitates EB1 tip tracking in cells. Finally, our results support a general principle in which microtubule-associated protein binding rates are influenced by the location and conformation of the protein binding interface.

# Materials and methods

**Key resources table**

| Reagent type (species) or resource | Designation | Source or reference | Identifiers | Additional information |
|---|---|---|---|---|
| Cell line (porcine, kidney) | LLC-Pk1 Express Tubulin-GFP | (*Rusan et al., 2001*) | RRID:CVCL_0391 | Gift from Dr. Pat Wadsworth |
| Cell line (porcine, kidney) | LLC-Pk1 Express EB1-GFP | (*Piehl and Cassimeris, 2003*) | RRID:CVCL_0391 | Gift from Dr. Lynne Cassimeris |
| Recombinant DNA reagent | pETEMM1-HIS6x-tev-EB1-GFP (plasmid) | (*Zanic et al., 2009*) | | |
| Recombinant DNA reagent | RP hk339-GFP (monomeric Kif5B-GFP-HIS6x plasmid) | (*Tomishige and Vale, 2000*) | Addgene plasmid #24431 | Gift from Dr. Ron Vale |

## Tubulin purification and labelling

Tubulin was purified and labeled as per *Gell et al. (2010)*.

## EB1-GFP purification

Plasmid pETEMM1-HIS6x-tev-EB1-GFP was transformed into Rosetta(DE3) pLysS *E. coli*, grown in 10 ml of LB+cam+kan at 37° overnight, and then subcultured into 1L of the same media, mixing at 37° for 2 hr to an A600 of 0.44. IPTG was then added to 2 mM and the culture was mixed at 18° for 14 hr. The culture was centrifuged 30 min. at 4°, 4400 xg. The cell pellet was resuspended into 12 ml of PBS/0.1% tween-20/5 mM B-mercaptoethanol/1 mg/ml lysozyme and protease inhibitors (1 mM AEBSF/10 µM pepstatin A/0.3 µM aprotinin/10 uM E-64), mixed at 4°, 2 hr., and then sonicated on ice at 80% power, 50% duty, 10 × 1 min. The lysate was centrifuged at 4°, 1 hr., 18000 xg. The supernatant was passed through 2 ml of Talon Metal Affinity Resin and the resin was sequentially washed with five column volumes Buffer A (50 mM sodium phosphate pH7.5/300 mM KCl/10% glycerol/5 mM B-mercaptoethanol). Next, the column was washed with 95% buffer A/5% bufferB (Buffer A/300 mM imidazole), followed by a wash with 90% A/10% B, and finally 85% Buffer A/15% Buffer B. All buffers had the protease inhibitors 1 mM AEBSF, 10 uM pepstatin A and 10 uM E-64). Protein was eluted with 1 ml fractions of 100% Buffer B + above protease inhibitors. Elution fractions were analyzed on coomassie and western blot. Relevant fractions were combined and dialyzed against Buffer A overnight at 4°, then 2 hr. at 18° in the presence of HIS6-taggedTEV protease (Expedeon TEV0010) at a 1:100 w/w protein ratio). Dialysate was then mixed with additional Talon resin to remove cleaved HIS6x tag.

## Kif5B-GFP purification

The RP hk339-GFP construct (monomeric Kif5B-GFP-HIS6x), a gift from Ron Vale (Addgene plasmid #24431; *Tomishige and Vale, 2000*), was transformed into the *E. coli* Rosetta (DE3) pLysS. This strain was cultured in LB+amp+cam at 30° overnight to an A600 of 1.5, then subcultured into 1200 ml of the same media, mixed at 37°, 3 hr to an A600 of 0.47. IPTG was added to 0.2 mM final and growth continued at 16° for 19 hr. The culture was centrifuged at 2100 xg, 4° for 30 min, supernatant discarded and cells resuspended in lysis buffer (50 mM sodium phosphate pH 6.2/250 mM NaCl/20 mM imidazole/5 mM B-mercaptoethanol/1 mM MgCl2/0.5 mM ATP/0.5% triton X-100 plus protease inhibitors (1 mM AEBSF/10 uM pepstatin A/10 uM E-64/0.3 µM aprotinin/1 mM benzamidine). Lysozyme was added to 0.1 mg/ml with 150U DNAse I and mixed at 4°, 1 hr. The cell suspension was lysed by sonication on ice, 2 × 10 min. at 90% power, 50% duty and centrifuged at 4°, 14,000 xg for 40 min. The supernatant was pH'ed to 7.6 and mixed with 1 ml Talon Metal affinity resin (Clontech 635509) for 30 min at 4°. Resin was washed 5x with two column volumes of lysis buffer + 0.1x protease inhibitors and protein was eluted with sequential 1 ml volumes of lysis buffer containing 300 mM imidazole pH7 + 0.1x protease inhibitors. The elution fractions were analyzed by coomassie and western blot and the peak fractions used in experiments.

## EB1 conjugation to gold beads

Unlabeled EB1 was conjugated to 20 nm gold particles using the Innovacoat Gold Particle labeling Mini Kit (Innovacoat 229–0005) as per kit instructions. EB1 was buffer-exchanged into PBS by centrifuging and washing with a 0.5 ml Amicon Ultra centrifugation filter (30 kd cut off). 10 µl (5.5 µg) of EB1 was diluted with 2 µl of kit dilution buffer and 42 µl of kit reaction buffer. 45 µl of this mixture was added to the kit's gold reagent and allowed to sit 25 min. 5 µl of the kit's quencher buffer was added and incubated for 5 min. 1 ml of a 1:10 dilution of quencher buffer was added, mixed, and Centrifuged at 4˚ for 20 min. at 9000 xg. The supernatant was discarded and the gold pellet was suspended in Brb80 buffer.

## Microtubule pool preparations

Microtubules for the bound nucleotides GDP, GMPCPP, and GTPγS of both closed and disrupted-structure were prepared as described in Reid et al. (2017).

Briefly, GDP microtubules were grown using a mixture of 33 µM tubulin, 1 mM GTP, 4 mM MgCl$_2$, and 4% DMSO, and incubated for 30 min at 37˚C. The solution was then diluted 100-fold into BRB80 solution with 10 µM Taxol and stored at either 37˚C (for closed microtubules) or at 25˚C (for disrupted-structure microtubules).

GMCPP microtubules were grown using a mixture of 3.9 µM tubulin and 1 mM GMPCPP in BRB80, which was incubated for 1 hr at 37˚C. To generate disrupted-structure GMPCPP microtubules, closed GMPCPP microtubules were incubated in 40 µM CaCl$_2$ for 40 min immediately before use in an experiment.

GTPγS microtubules were grown using a mixture of 12 µM tubulin, 50 mM KCl, 10 mM DTT, 0.1 mg/ml Casein, 4 mM GTPγS, and unlabeled GMPCPP 'seed' microtubules to serve as nucleation points. The mixture was incubated for 1 hr at 37˚C, then diluted 3.5-fold and stored at 37˚C overnight. The disrupted-structure GTPγS microtubules were prepared similarly, but with the following changes; the initial mixture contained higher tubulin concentration (25.5 µM instead of 12 µM), after incubation the mixture was diluted 26-fold instead of 3.5-fold, and was stored at 25˚C overnight instead of at 37˚C.

## Construction and preparation of flow chambers for TIRF microscopy imaging

Imaging flow chambers were constructed as in Section VII of Gell et al. (2010), with the following modifications: two narrow strips of parafilm replaced double-sided scotch tape as chamber dividers: following placement of the smaller coverslip onto the parafilm strips, the chamber was heated to melt the parafilm and create a seal between the coverslips; typically only three strips of parafilm were used, resulting in two chambers per holder. Chambers were prepared with anti-rhodamine antibody followed by blocking with Pluronic F127, as described in Section VIII of Gell et al. (2010).

Microtubules were adhered to the chamber coverslip, and the chamber was flushed gently with warm BRB80. The flow chamber was heated to 35˚C using an objective heater on the microtubule stage, and then 3–4 channel volumes of imaging buffer were flushed through the chamber.

Microtubules were imaged on a Nikon TiE microscope using 488 nm and 561 nm lasers sent through a Ti-TIRF-PAU for Total Internal Reflectance Flourescence (TIRF) illumination. An Andor iXon3 EM-CCD camera fitted with a 2.5X projection lens was used to capture images with high signal to noise and small pixel size (64 nm). Images were collected using TIRF with a Nikon CFI Apochromat 100 × 1.49 NA oil objective.

## EB1-GFP binding to GMPCPP microtubules

Closed GMPCPP microtubules, prepared as described above, were introduced into an imaging chamber and allowed 30 s-3min to bind the antibody on the coverslip. The imaging chamber was then flushed with 1X Imaging Buffer. 1X Imaging Buffer consisted of BRB80 supplemented with the following: Casien 80 µg/ml, D-Glucose 20 mM, Glucose oxidase 20 µg/ml, Catalase 10 µg/ml, DTT 10 mM, KCl 30 mM, and Tween-20 1%.

EB1-GFP solution (100 µL Imaging Buffer, 1.5 µL of 11 µM EB1-GFP (163 nM final EB1-GFP concentration) was then introduced into the imaging chamber and allowed 10–15 min to bind, after

which images were collected in both the microtubule (red) and EB1-GFP (green) channels (561 nm and 488 nm respectively).

## EB1-GFP binding to microtubule pools

GDP microtubules, either closed or disrupted-structure, were prepared as described above and introduced into an imaging chamber. Microtubules were allowed 30 s-3min to bind the antibody on the coverslip. The microtubule solution was then flushed out of the chamber with 1X imaging buffer (2X Imaging buffer = 110 mM KCl, 40 µg/ml Glucose Oxidase, 20 µg/ml Catalase, 40 mM D-Glucose, 20 mM DTT, 160 µg/ml Casein, 2% Tween-20, and 20 µM Taxol). EB1-GFP solution (50 µL 2X imaging buffer, 45 uL BRB80, 5 µL of 5 µM EB1-GFP) was flowed into the imaging chamber and allowed 10–15 min to bind, after which images were collected in both the microtubule (red) and EB1-GFP (green) channels (561 nm and 488 nm, respectively).

GMPCPP microtubules, either closed or disrupted-structure, were prepared as described and introduced into an imaging chamber. Microtubules were allowed 30 s-3min to bind the antibody on the coverslip. The microtubule solution was then flushed with low-salt imaging buffer (Low-salt 2X Imaging buffer = 25 mM KCl, 40 µg/ml Glucose Oxidase, 20 µg/ml Catalase, 40 mM D-Glucose, 20 mM DTT, 160 µg/ml Casein, 2% Tween-20, and 20 µM Taxol). EB1-GFP solution (50 µL low-salt 2X imaging buffer, 45 uL BRB80, 5 µL of 5 µM EB1-GFP) was flowed into the imaging chamber and allowed 10–15 min to bind, after which images were collected in both the microtubule (red) and EB1-GFP (green) channels (561 nm and 488 nm, respectively).

GTPγS microtubules, either closed or disrupted-structure, were prepared as described and introduced into an imaging chamber. Microtubules were allowed 30 s-3min to bind the antibody on the coverslip. The imaging chamber was then flushed with 1X imaging buffer (2X Imaging buffer = 110 mM KCl, 40 µg/ml Glucose Oxidase, 20 µg/ml Catalase, 40 mM D-Glucose, 20 mM DTT, 160 µg/ml Casein, 2% Tween-20, and 20 µM Taxol). EB1-GFP solution (50 µL 2X imaging buffer, 45 µL BRB80, 5 µL of 5 µM EB1-GFP (250 nM final EB1-GFP concentration)) was flowed into the imaging chamber and allowed 10–15 min to bind, after which images were collected in both the microtubule (red) and EB1-GFP (green) channels (561 nm and 488 nm, respectively).

## EB1-GFP on-rate and dwell time movies

Microtubules were prepared as described above and introduced into an imaging chamber. Microtubules were allowed 30 s-3 min to bind the antibody on the coverslip. The imaging chamber was then flushed with 1X imaging buffer (2X Imaging buffer = 110 mM KCl, 40 µg/ml Glucose Oxidase, 20 µg/ml Catalase, 40 mM D-Glucose, 20 mM DTT, 160 µg/ml Casein, 2% Tween-20, and 20 µM Taxol). EB1-GFP solution (25 µL 2X imaging buffer, 20 µL BRB80, 5 µL of 5 µM EB1-GFP) was flowed into the imaging chamber. Using a capture rate of 100 frames per second, 10 s of images were collected per microtubule in both the microtubule (red) and EB1-GFP (green) channels (561 nm and 488 nm, respectively).

## Kif5B-GFP global binding to microtubules

GDP microtubules, either closed or disrupted-structure, were prepared as described above and then introduced into an imaging chamber. Microtubules were allowed 30 s-3min to bind the antibody on the coverslip. The microtubule solution was then flushed with 1X imaging buffer (2X Imaging buffer = 110 mM KCl, 40 µg/ml Glucose Oxidase, 20 µg/ml Catalase, 40 mM D-Glucose, 20 mM DTT, 160 µg/ml Casein, 2% Tween-20, and 20 µM Taxol). Either 2.5 nM Kib5B-GFP solution (25 µL 2X imaging buffer, 14.5 uL BRB80, 5 µL of 25 µM Kif5, and 5 µL 100 mM AMPPNP) or 10 nM Kif5B-GFP solution (25 µL 2X imaging buffer, 14.5 uL BRB80, 5 µL of 100 µM Kif5, and 5 µL 100 mM AMPPNP) was flowed into the imaging chamber and allowed 10–15 min to bind, after which images were collected in both the microtubule (red) and Kif5 (green) channels (561 nm and 488 nm, respectively).

## EB1-GFP binding to GDP and GDP+P$_i$microtubules

GDP microtubules, either closed or disrupted-structure (prepared as described above), were introduced into an imaging chamber, as described above. The imaging chamber with GDP microtubules was then flushed with either 1X GDP+P$_i$ imaging buffer (2X GDP+P$_i$ Imaging buffer = 110 mM

$KPO_4$, 40 µg/ml Glucose Oxidase, 20 µg/ml Catalase, 40 mM D-Glucose, 20 mM DTT, 160 µg/ml Casein, 2% Tween-20, and 20 µM Taxol; all in Brb80) or 1X GDP imaging buffer (2X GDP imaging buffer = 110 mM KCl, 40 µg/ml Glucose Oxidase, 20 µg/ml Catalase, 40 mM D-Glucose, 20 mM DTT, 160 µg/ml Casein, 2% Tween-20, and 20 µM Taxol; all in Brb80), and were allowed to sit for 20 min. Microtubules were flushed with 1X imaging buffer again, after which an EB1-GFP solution (20 µL 2X GDP+$P_i$ or GDP imaging buffer, 13 µL BRB80, 4 µL of 2.5 µM EB1-GFP) was flowed into the imaging chamber. EB1-GFP was allowed 10 min to bind, after which images were collected in both the microtubule (red) and EB1-GFP (green) channels (561 nm and 488 nm, respectively).

## EB1-GFP binding to GDP and GDP-$P_i$ analogue ($AlF_4^-$) microtubules

Stock solutions of 1 mM $AlCl_3$ and 100 mM NaF were diluted into the imaging buffers to the concentrations shown. Together, the $Al^{3+}$ and $F^-$ ions form the Pi analogue $AlF_4^-$ that mimics the phosphate group with 1000-fold higher affinity (*Antonny and Chabre, 1992*; *Carlier et al., 1988*; *Petsko, 2000*).

Closed GDP microtubules (prepared as described above), were introduced into an imaging chamber, as described above. The imaging chamber with GDP microtubules was then flushed with either 1X GDP-$P_i$-Analog imaging buffer (2X GDP-$P_i$-Analog Imaging buffer = 105 mM KCl, 4 mM NaF, 20 µM $AlCl_3$, 40 µg/ml Glucose Oxidase, 20 µg/ml Catalase, 40 mM D-Glucose, 20 mM DTT, 160 µg/ml Casein, 2% Tween-20, and 20 µM Taxol; all in Brb80) or 1X GDP imaging buffer (2X GDP imaging buffer = 105 mM KCl, 4 mM NaF, 40 µg/ml Glucose Oxidase, 20 µg/ml Catalase, 40 mM D-Glucose, 20 mM DTT, 160 µg/ml Casein, 2% Tween-20, and 20 µM Taxol; all in Brb80), and were allowed to sit for 20 min. Microtubules were flushed with 1X imaging buffer again, after which an EB1-GFP solution (2X GDP-$P_i$-Analog or GDP imaging buffer, BRB80, EB1-GFP) was flowed into the imaging chamber. EB1-GFP was allowed 10 min to bind, after which images were collected in both the microtubule (red) and EB1-GFP (green) channels (561 nm and 488 nm respectively).

## Cell lines

The LLC-Pk1 cell line expressing EB1-GFP was a gift from Patricia Wadsworth, and the cell line expressing GFP-Tubulin was a gift from Lynne Cassimeris. The identity of the cell lines (non-human) were authenticated by microscopy observation and analysis. To test the LLC-Pk1 strains containing EB1-GFP and Tubulin-GFP for mycoplasma contamination, we used the Sigma-Aldrich LookOut Mycoplasma PCR Detection Kit (#MP0035) along with Sigma-Aldrich JumpStart Taq Polymerase (#D6558) as per the kit instructions. The test was negative for mycoplasma contamination for both strains.

## Culture, imaging, and drug treatment of LLC-Pk1 cells

The LLC-Pk1 cell line expressing EB1-GFP, or expressing GFP-Tubulin, was grown in Optimem media (ThermoFisher #31985070), 10% fetal bovine sera + penicillin/streptomycin at 37° and 5% $CO_2$. Cells were grown in 35-mm glass bottom dishes for visualization by microscopy.

For drug treatments, the Optimem media was replaced with $CO_2$ independent imaging media and drug dissolved in DMSO, or only DMSO as a control, was added to the final concentration needed. Cells were incubated with the drug/DMSO for 30 min prior to imaging.

## Analysis of EB1-GFP binding profile on GMPCPP microtubules

Microtubule and EB1-GFP fluorescence intensity were analyzed, and the microtubule aligned, as previously described (*Coombes et al., 2016*). Briefly, the single time point images of GMPCPP microtubules with EB1-GFP were cropped to separate each microtubule into a single image using ImageJ. Then, integrated and averaged line scans of each microtubule image were created using a MATLAB script. In each case, the microtubule was aligned with the brighter EB1-GFP (green) fluorescence end on the right, and then the red and green fluorescence were plotted as a function of microtubule length from the dimmer EB1-GFP signal end to the brighter EB1-GFP signal end. To do this, the green EB1-GFP fluorescence was integrated along the length of each microtubule ± 4 pixels above and below the microtubule centerline to account for point spread function and variability in properly finding the microtubule centerline. Then, the green fluorescence intensity was summed over the last 9 pixels on both ends of the microtubule. The lower summed value was considered the dimmer EB1-

GFP end, while the higher summed value was deemed the brighter EB1-GFP end. To combine all individual microtubule data into an ensemble average plot, the microtubules were rebinned to a common length, represented by the mean length of all observed microtubules (*Gardner et al., 2005*). Scatter plots of the ensemble average values were created by importing the integrated line scan fluorescence data into Excel.

## Microtubule tip structure analysis

Average tip standard deviation values were calculated by fitting an error function to the microtubule ends, as previously described (*Demchouk et al., 2011*).

## EB1-GFP/tubulin intensity ratio analysis

The EB1-GFP/Tubulin intensity ratio was analyzed for every individual microtubule using a previously described MATLAB script (*Coombes et al., 2016*). Briefly, the ends of each microtubule image were manually clicked by a user. Then, the average green and red fluorescence intensity between the clicks, including +/- 4 pixels above and below the centerline (256 nm) established by the two clicks, was calculated. Background in the red and green channels was estimated by calculating the average green and red fluorescence between the two clicks, but in the pixels 8–22 above the centerline established by the two clicks, and in the pixels 8–22 below the centerline established by the two clicks. For each microtubule, the average EB1-GFP signal over background was calculated by dividing the average signal along the microtubule by the average background estimate. Similarly, the average red microtubule signal over background was calculated by dividing the average red microtubule signal by the average red background estimate. Finally, the intensity ratio of average green signal over background to average red signal over background was calculated and reported for each microtubule.

## EB1-GFP binding coverage analysis

To compare relative binding of EB1-GFP on GDP and GDP-$P_i$ microtubules, and in the presence of Vinblastine, the total length of green (EB1-GFP) occupancy was divided by the total length of the red microtubules on each image (defined as EB1-GFP 'microtubule coverage'). This was accomplished by using previously described semi-automated MATLAB analysis code (*Reid et al., 2017*). Briefly, first, automatic processing of the red microtubule channel was used to determine the microtubule-positive regions, which then allowed for conversion of the red channel into a binary image with white microtubules and a black background. The green EB1-GFP channel was then also preprocessed to smooth high-frequency noise and to correct for TIRF illumination inhomogeneity. The green channel threshold was then manually adjusted to ensure visualization of all EB1-GFP binding areas on each microtubule. Measurements of the total EB1-GDP length were then automatically collected from the identified microtubule regions. For presentation, the EB1-GFP microtubule coverage values for each image and condition were normalized to a specific grand average control value.

## Analysis of peak EB1-GFP binding to microtubules in LLC-Pk1 cells

To analyze the peak EB1-GFP binding to growing microtubule ends, individual comets were analyzed using MATLAB. EB1-GFP comet movies were first analyzed to ensure that only mature comets were selected for analysis. For each mature comet, MATLAB code was used which allowed two manual clicks on each comet: at the location of the highest intensity spot on each comet, and on the microtubule lattice just outside the end of the comet. In addition, the user clicked twice to measure the length of the comet. Then, the fluorescence intensity at the highest intensity spot on the comet was divided by the fluorescence intensity on the microtubule lattice outside of the comet, which we defined as the normalized peak EB1-GFP tip fluorescence.

## Electron microscopy experiments

To collect electron microscopy images of GMPCPP microtubules, the microtubules were prepared as described above. One drop of the microtubule solution was placed on a 300-mesh carbon coated copper grid for a duration of 60 s, after which the grid was stained with 1% uranyl acetate for 60 s. The stain was the wicked away using filter paper, and the grid was left to dry and then stored until use. The samples were imaged using an FEI Technai Spirit BioTWIN transmission electron

microscope. Cryo-EM was performed on the same electron microscope. Samples were prepared on a 300-mesh copper grid with a lacey-carbon support film. Grids were treated in a Pelco Glow Discharger before the addition of the GDP microtubule sample and freezing in vitreous ice using a FEI Vitrobot.

Electron microscopy imaging of EB1 conjugated to gold beads was performed by placing one drop of a solution containing disrupted-structure GDP microtubules (prepared as described above) with gold-bead-conjugated EB1 onto a 300-mesh carbon coated copper grid. The sample grid was treated identically to the procedure described above for TEM of GMPCPP microtubules.

## Analysis of EB1-bead experiment

Images of EB1 conjugated to gold beads were analyzed in two parts. First, each image was examined manually. Gold beads appearing in an image were tallied and classified according to their proximity to a microtubule and the appearance of the microtubule at the location of the bead. Only beads directly contacting a microtubule were tallied, and were classified into the following categories: Edges, for gold beads localized at the edges of sheet-like regions of a microtubule or at a microtubule end; Defect, for gold beads localized to regions of the microtubule with visible defects such as gaps or breaks; and Lattice, for gold beads localized to intact regions of the microtubule. Additionally, gold beads found on sheet-like regions were sub-classified based on their location on the sheet, as either Sheet-Edge or Sheet-Middle. To be classified as a Sheet-Edge, a bead was required to overlap with the sheet edge but have its center located beyond, not overlapping, the sheet. Secondly, using a previously described MATLAB script, the total perimeter of the all microtubules in the gathered images was traced and each segment was classified as being Lattice or Edge/Defect, based on how a bead would have been classified if it was located at the given segment of microtubule (*Reid et al., 2017*). The lengths for each classification were summed and used to normalize the counts.

## EB1 3D diffusion and binding simulation: setup and assumptions

Approximations for the shapes of a tubulin dimer and EB1 microtubule binding domain (PDB ID: 3JAR; *Zhang et al., 2015*) were created manually based off of Cryo-EM reconstruction data using 3D modeling software (Blender) and exported to an. obj object file format. EB1's binding site coordinates were determined by calculating the center of the contacting faces for each of the four adjacent tubulin dimers, and the binding center was determined as the average position of the four binding interface coordinates.

The simulation was coded and run using MATLAB 2015a (Mathworks). The tubulin dimer subunits were arranged into the canonical microtubule arrangement, with either a blunt or tapered end, maintaining 207 tubulin dimers (average protofilament length ~16 dimers) for both conditions. EB1 was initialized at a random position on a sphere of radius 500 nm centered on the microtubule. We note that while the simulated microtubule shape was approximately cylindrical with a height of ~128 nm, the random initial localization of EB1 within a 500 nm radius of the microtubule did not bias the simulated EB1 molecules toward the microtubule ends, as demonstrated by our face-binding simulations (*Figure 4F*), in which no end bias was observed.

## EB1 3D diffusion and binding simulation: dynamic calculation of time step size

In order to run the simulation efficiently, the time step for the simulation was dynamically adjusted based on the distance of EB1 from the microtubule, similar to *Castle and Odde (2013)*. Briefly, the time step was large (~$7 \times 10^{-6}$ s) when the simulated EB1 molecule was far from the microtubule ($\geq$100 nm), and small (~$4 \times 10^{-12}$ s) when the EB1 molecule was close to the microtubule ($\leq$1 nm). Specifically, the scaled time step sizes were calculated based on the translational diffusion speed of the EB1 molecule, where the translational diffusion step size was governed by:

$$<\Delta x^2> = 2dD_c\Delta t \qquad (1)$$

Where $\Delta x$ = translational diffusion molecule step size, $d$ = dimensions (=3), $D_c$ = translational diffusion coefficient (see below), and $\Delta t$ = time step size. This equation was then used to determine the scaled time step size, such that when the EB1 molecule was $\geq$100 nm from the microtubule (e.g.,

$Dist_i \geq 100$), the time step size was then calculated from *Equation 1* such that the molecule would be five steps away from the microtubule (e.g., $\Delta x_i \equiv Dist_i/5$), and when EB1 was $\leq 1$ nm from the microtubule, the time step size was then calculated from *Equation 1* such that the molecule would be 20 steps away from the microtubule (e.g., $\Delta x_i \equiv Dist_i/20$). The time step size was scaled linearly between those lengths, and so that a step closer to the microtubule reduced the time step size, until the EB1 molecule was 1 nm from a binding site, at which time a fixed time step was used that resulted in typical diffusion lengths of 0.5 angstroms per step. The distance used in these calculations ($Dist_i$) was the separation distance between the closest points on the microtubule and on EB1, not the center-to-center distance.

## EB1 3D diffusion and binding simulation: EB1 3D diffusion

Each EB1 molecule was allowed to diffuse both translationally and rotationally until the molecule was either further than 2000 nm from the simulation center, or else had the EB1-binding interface properly oriented and within 1 nm of the binding interface center of any microtubule binding location. EB1 diffusion was governed by the calculated translational and rotational diffusion coefficients as determined by *Equations 2 and 3*, respectively:

$$D_c = k_b T / 6\pi\rho r \tag{2}$$

$$D_{c\_rot} = k_b T / 8\pi\rho r^3 \tag{3}$$

In the above equations, ρ is viscosity of water, and r is the radius of the EB1 molecule, approximated as the average radius of all its vertex points. Diffusion and rotation was assumed to be equal across all axes.

The translational distance that the EB1 molecule traveled in a time step was then calculated based on random sampling from a Gaussian distribution governed by:

$$d_x = r_{Gauss}\sqrt{2D_c\Delta t} \tag{4}$$

$$d_y = r_{Gauss}\sqrt{2D_c\Delta t} \tag{5}$$

$$d_z = r_{Gauss}\sqrt{2D_c\Delta t} \tag{6}$$

where $d_x$, $d_y$, and $d_z$ were the translational displacement in x, y and z respectively, and $r_{Gauss}$ was a normally distributed random number (mean: 0, standard deviation: 1).

Similarly, in the same time step, the angle of rotation for EB1 was also determined from normally distributed random number, and the rotational diffusion constant:

$$rot_x = r_{Gauss}D_{c\_rot}\Delta t \tag{7}$$

$$rot_y = r_{Gauss}D_{c\_rot}\Delta t \tag{8}$$

$$rot_z = r_{Gauss}D_{c\_rot}\Delta t \tag{9}$$

## EB1 3D diffusion and binding simulation: EB1 collision with a microtubule

EB1 was not permitted to pass through or intersect volumes with any tubulin dimers. The dynamic time steps prevent skipping from one side of a tubulin dimer to the other in one time step, thus avoiding collision, by maintaining appropriately small step sizes when near a tubulin dimer. Collisions and measurements of the nearest point distance between EB1 and any tubulin subunit were conducted using the Voronoi-Clip (vClip) algorithm (*Mirtich, 1998*).

Briefly, the closest point between the EB1 polyhedron and a tubulin dimer polyhedron were determined by randomly selecting initial candidate features (vertex, edge or face) for each object. Then, the neighboring features of the EB1 candidate were tested to find one that was closer to the Tubulin candidate feature. If a feature was found to be closer than the initial candidate, it became

the new candidate. This was repeated for both the EB1 and Tubulin objects until no two candidates could be found that were closer together than the current pair. Then, if the distance between them was less than or equal to zero (with respect to the object's outer surface), that indicated that they were intersecting and a collision had occurred, in which case the EB1 position was reverted to the previously recorded value, and the EB1 molecule permitted another attempt at diffusion without collision. If the distance between candidate features was greater than zero, no collision had occurred and the simulation proceeded to the next time step.

The vClip method is useful in reducing computational load, because the final candidate features can be stored from the last time step and used as the initial candidate features for the next test (*Mirtich, 1998*). Because the time steps are small, the saved candidate features are often the optimal candidate features (or adjacent to the optimal candidate features) for the new position (that has moved and rotated only slightly).

A second optimization technique was used to avoid testing against every tubulin dimer at every time step. Briefly, a much more rapid test for center to center distance was calculated for EB1 to detect 'nearby' tubulin dimers (determined by the sum of the maximum vertex radius for both EB1 and the tubulin dimer, and a buffer distance) as compared to the more expensive and accurate vClip test. 'Nearby' was determined by a 'Sweep and Prune' algorithm, specifically a hierarchical binary tree of tubulin dimer bounding-boxes that was used to very quickly determine if any tubulin dimers needed be considered 'Nearby'.

## EB1 3D diffusion and binding simulation: EB1 binding to a microtubule

EB1 stopped diffusing, and the simulation was ended for that molecule, when its binding interface was *oriented properly* and within 1 nm of a binding interface on the microtubule. This 'binding' configuration was constrained by both distance and molecule orientation relative to its binding site on the microtubule: if the binding interface was oriented *away* from the microtubule, the separation distance could be as little as 0 nm, and the binding interface would still be >4 nm (EB1 diameter) away from the tubulin pocket binding interface. In this situation, the simulation would not stop, but the EB1 molecule would continue to diffuse and rotate until it entered into the proper binding position, or diffused out of the simulated volume.

## Acknowledgements

This work was supported by National Institutes of Health grant NIGMS R01-GM103833 and R35-GM126974 and National Science Foundation CAREER award 1350741 to MKG. MZ acknowledges the support of National Institutes of Health grant R35-GM119552, the Human Frontier Science Program and the Searle Scholars Program. The authors have no conflicts of interest to disclose. Parts of this work were carried out in the Characterization Facility, University of Minnesota, a member of the NSF-funded Materials Research Facilities Network (www.mrfn.org) via the MRSEC program. We thank Drs. Joe Howard and Ron Vale for gifting reagents used as part of this project. We thank Holly Goodson and the Gardner Laboratory for helpful discussions.

## Additional information

### Funding

| Funder | Grant reference number | Author |
| --- | --- | --- |
| National Institutes of Health | R01-GM103833 | Melissa K Gardner |
| National Institutes of Health | R35-GM126974 | Melissa K Gardner |
| National Science Foundation | 1350741 | Melissa K Gardner |

The funders had no role in study design, data collection and interpretation, or the decision to submit the work for publication.

## Author contributions
Taylor A Reid, Conceptualization, Software, Formal analysis, Validation, Investigation, Visualization, Methodology, Writing—original draft, Writing—review and editing; Courtney Coombes, Formal analysis, Investigation, Methodology; Soumya Mukherjee, Rebecca R Goldblum, Kyle White, Sneha Parmar, Marija Zanic, Investigation; Mark McClellan, Validation, Investigation, Methodology; Naomi Courtemanche, Conceptualization; Melissa K Gardner, Conceptualization, Resources, Data curation, Software, Formal analysis, Supervision, Funding acquisition, Methodology, Project administration, Writing—review and editing

## Author ORCIDs
Marija Zanic (iD) http://orcid.org/0000-0002-5127-5819
Melissa K Gardner (iD) https://orcid.org/0000-0001-5906-7363

## Decision letter and Author response
Decision letter https://doi.org/10.7554/eLife.48117.014
Author response https://doi.org/10.7554/eLife.48117.015

# Additional files

## Supplementary files
• Transparent reporting form
DOI: https://doi.org/10.7554/eLife.48117.012

## Data availability
All data generated or analysed during this study are included in the manuscript and supporting files.

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
