## [Decision Letter]

[Editors’ note: a previous version of this study was rejected after peer review, but the authors submitted for reconsideration. The first decision letter after peer review is shown below.]

Thank you for submitting your work entitled "Microtubule Structural State Recognition Can Explain the Rapid Flux of EB1 to Microtubule Ends" for consideration by *eLife*. Your article has been reviewed by a Reviewing Editor and a Senior Editor, a Reviewing Editor, and three reviewers. The following individuals involved in review of your submission have agreed to reveal their identity: Manuel Thery (Reviewer #1).

Our decision has been reached after consultation between the reviewers. Based on these discussions and the individual reviews below, we regret to inform you that your work will not be considered further for publication in *eLife*.

A major point of interest in the manuscript is the role of specific microtubule lattice structures (defects/edges) in the initial binding of EB1 to microtubules. One reviewer considered that this, and the implication that nucleotide state of tubulin is not the key feature for recognition, was intrinsically interesting. The other two reviewers felt that you were discussing a more nuanced scenario where partial/edge binding sites provide the first binding interaction for EB1 which is then locked in by arrival of new tubulins. These two reviewers were, however, not convinced that the data in the paper were sufficient to support this model over other possibilities.

We discussed the paper among the reviewers at some length and I am afraid the consensus was that it is not possible to ask for revisions at this stage. I include all the reviewers comments below so you can get an idea of whether you want to try a new submission to *eLife* or to modify the manuscript to submit elsewhere.

Reviewer #1:

In this manuscript the authors argue that EB1 recognizes specific microtubule lattice structures (presenting the lateral side of tubulin dimers) rather than the nucleotide state of the tubulin (i.e. GTP versus GDP). Thereby they contradict a long-standing belief in the field. As such this paper is ground breaking and deserves publication in a top quality journal such as *eLife*.

The main arguments allowing them to reach these conclusions are:

- EB1 binds more frequently to the plus tips of GMPCPP microtubules (ie microtubules made of GTP-like tubulin) than along the lattice. So the tubulin state is not sufficient to determine the affinity of EB1.

- EB1 binds the transition between microtubule seeds and the GMPCPP lattice even more than plus tip. So it is not the tip per se but some structural features that EB1 binds to.

- EB1 binds disrupted lattice more than intact lattice. They used EB1-beads and cryo EM to correlate lattice structure and EB1 localization.

- EB1 binds the edge of sheet-like microtubule structures rather than the central part of the sheet. Arguing again that it recognizes a structure more than the nucleotide state.

- key control experiment: Kif5b, which is supposed to recognize the outer rather than the lateral part of tubulins (is this really clear? Demonstrated previously?), does not bind disrupted lattice more than intact lattice.

My concerns are the following:

- differences between intact lattice and tips on GDP microtubules are significant but really small. Differences are huge when the lattice is so much damaged that is incompatible with actual microtubules (they would depolymerise immediately). Have these difference any physiological relevance?

- tricks to disrupt the lattice are quite strong and may have unexpected side-effects.

- On dynamic GDP/GTP microtubules there is a high concentration of EB1 at the growing tips. Does the nucleotide state also matters? Or only the structure? Does the normalization in Figure 2C (between normal and disrupted lattices) hide the fact that EB1 binds much more to GMPCPP and GTPgammaS microtubules than to GDP microtubule and thus that the tubulin state actually matters "more" than the lattice structure? It would be nice to show the non-normalized values and comment on this point.

- The fact that EB1 binds more to tapered end than to the lattice is known (Guesdon et al, 2016). The fact that EB1 binds to the side of tubulin dimers is known (Zhang et al, 2015). The fact that Kif5b binds the outer part of tubulin is known (at least it is assumed by the authors but I am not sure how this is actually demonstrated). So, it is no surprise that EB1 binding is structure-sensitive and that Kif5b is not. One could consider that all evidences were already present and thus contest the actual innovation of this work.

- I can't prevent myself from being frustrated that our work, which showed for the very first time some tubulin dynamics on supposedly damaged lattice regions is not cited here. I hope this is not intentional to claim for more novelty.

- The authors did not discuss the work of Pous and Perez, 2016 in which they showed that Clip170 binds to damaged microtubule lattice. We should also take into account that the observations in Pous and Perez paper somehow limit the novelty of this work.

Reviewer #2:

The submission from Reid et al., addresses an interesting question about how EB1 recognizes the growing microtubule end. Although there has been a lot of work on this question from multiple groups, I think it is fair to say that the science is not exactly settled. In their work, the authors use quantitative fluorescence microscopy along with computational simulations of association and methods developed by the Gardner lab to create microtubules with 'damaged' regions (disruptions in the lattice). The logic of the manuscript goes as follows. Tow experimental observations (i) that EB1 prefers to bind more tapered microtubule ends, which correlates with the plus end; EB1 also localizes preferentially at the transition between seed and new growth, which may contain lattice defects (Figure 1); and (ii) that EB1 prefers to bind microtubules containing disruptions (Figure 2); motivate the hypothesis that EB1 may prefer to associate with 'edge' sites. An implementation of this hypothesis into a computational simulation of association kinetics shows that edge association can lead to faster association (Figure 3), and control simulations using a 'face binder' (and a control experiment using kinesin) demonstrate that edge binding is not an obligatory outcome of this kind of modeling. This leads to a model (Figure 5) wherein EB1 arrives via a 'lock-in' mechanism that entails rapid (but weak/transient) associations with partial/edge-binding sites that can get 'locked in' to a higher affinity state by arrival of new ab-tubulins that complete the EB1 binding site.

I appreciate that the authors are looking at this question in a new way, and deploying new approaches to do it. The manuscript is well-written, and the figures are logical and clear and obviously prepared with care. However, there appear to be some inconsistencies in logic/interpretation that make me question whether all of the data add up in the way the authors say they do. Additionally, and independent of my concern about inconsistency, given that this is a pretty well-studied problem already, I don't think that in its present form the manuscript has done enough to convincingly rule out alternative mechanisms (e.g. that EB simply associates directly with 'cap' sites without going through a 'lock-in'), to justify the need for the alternative model presented, or to show it can quantitatively explain existing data from other groups (e.g. Maurer et al., 2014). In the end, while it may be that EB1 can rapidly and transiently associate with edge sites, I'm not yet convinced that is necessary for the mechanism of its end binding.

Connection between experiments and simulations.

The experiments showing association of EB1 with damaged lattices are interesting. But the molecular nature of those sites is not known. The authors pursue a model in which the defining characteristic of those sites is elevated edge content. But can they rule out that in those damaged sites tubulin adopts more 'sheet-like' conformations that mimic end taper and curvature (e.g. Guesdon)?

- They simulate association kinetics, but nearly all experiments measure binding in a way that probably reflects affinity

The simulations (for which the use of blunt and tapered microtubule segments is clever) simulate the rate of association in what I think is a reasonable way. Mostly they are clear about this, but in a few places the work 'binding' is used in a way that could be misleading. The authors reasonably state that these partial edge binding sites are likely to be low-affinity ones, but multiple supporting figure panels essentially reflect equilibrium binding, which should show the high-affinity sites, not the low-affinity ones the authors care about. This apparent contradiction needs to be clarified. I think the details of the 'lock-in' also need to be fleshed out in terms of the relative kinetics of EB1 dissociation from edge sites and ab-tubulin association to complete EB1 binding sites. Under what regimes can this work?

- Integration with current knowledge

Maurer et al., 2014 contains a minimal kinetic model that can describe quantitative observations about EB1 comets. The authors are essentially proposing a different model for how EB1 gets to its comet site. I think they need to explain better why an alternate model is needed, and maybe even show that theirs can actually do what Surrey's toy model did.

In subsection “Rapid frame-rate fluorescence microscopy experiments demonstrate increased on-rate of EB1-GFP to disrupted-structure microtubule pools”, the wording seems to indicate that Guesdon et al., also observed significant edge binding. But if I read Guesdon et al., correctly, only 2.1% of their observed binding was at an edge or on the inside of a sheet. This is a borderline minor point, but I think they authors need to be clearer about where prior data support their model, and about where their data appear to be in conflict with prior data.

Reviewer #3:

Reid et al. have studied how EB1-GFP binds to different microtubule structures using TIRF microscopy, electron microscopy, and computational modeling. They show that EB1-GFP binds to the ends of GMPCPP microtubules at intermediate salt concentrations and binds preferentially to "disrupted" microtubules that were prepared with Taxol. The idea is that these structures have incomplete EB binding sites, which is why EB prefers them. Their computational model is a Brownian dynamics simulation that measures the amount of time that EB diffuses become coming close to its binding site. They propose that EB binds with faster on-rates to incomplete binding sites located at the very end of the microtubule. They conclude with a model for EB end-recognition wherein the tapered, incomplete end of the lattice is the location where EB loads onto the microtubule.

It's important to state that this manuscript is squarely at odds with some ideas circulating in the field. A few labs (e.g, Surrey) have published data showing that the EB signal is displaced back from the very end of the microtubule, with polymerases like XMAP215 being found at the very end instead. There are a few hypotheses for why EB is displaced. One hypothesis is that EB prefers GDP-Pi lattices (Nogales). My favorite idea is that the end is tapered, which means that the binding sites for EB1 are incomplete, and thus EB doesn't bind there. I should say that a recent paper by Guesdon et al., used EM to show that EB can bind at the very end, but the fluorescence data is very hard to ignore, given the much larger number of observations relative to the EM. This manuscript argues that the "EB-free zone" is not EB free and is rather the zone that EB uses to load onto the lattice.

Overall, the manuscript does a poor job engaging with the debate. Also, in order to overturn an emerging consensus, the expectation re: data and analysis is high. Unfortunately, I'm not sure that the paper brings enough to the table. I will highlight some technical concerns that I have below. But I'm surprised to have a paper on EB binding that doesn't have any analysis of end-tracking of dynamic microtubules in it. There is no direct evidence that the tapered end is where the EB loads on, which could only come from clever experiments using dynamic microtubules. Combined with the problems highlighted below, I cannot support publication in *eLife*.

Figure 1

1) I would like to see some positive control data on their EB1-GFP. Does the protein form nice comets that match the literature in terms of size, etc.? Does the salt concentration they use affect end-tracking behavior? Are the EB1-GFP signals in Figure 1B and Figure 1F single molecules? They look like larger puncta. If they are not single molecules, are the larger signals forming by accumulation or are they aggregates? Quality control of this type would be very welcome.

2) I was surprised by the observation of extensive tapering on the ends of the GMPCPP microtubules in Figure 1C. Where does this tapering coming from? The degree of tapering observed by TIRF microscopy (on the order of 1.5 um) is much larger than the degree of tapering observed by EM, where the length of the taper is similar to the width of the MT (on the order of 25 nm). Why does the degree of tapering differ so much? Microtubule ends are sometimes ragged after they are broken during pipetting. But even for GMPCPP microtubules, these ragged ends shrink back to blunt ends within a few minutes. So my concern is that the tapering in Figure 1C is coming from the computational procedure wherein the microtubules were aligned and then rebinned to a common length. If the end of each microtubule is not at the same location within the bins, this will create a spreading out of the fluorescence signal, won't it? The methods did not provide enough information about the alignment and binning procedure for me to rule this out. The methods did make clear that the fitting of the data to the error function happened only after the alignment and rebinning. Individual microtubules were not fit to the error function, which means that the alignment procedure did not have sub-pixel accuracy. Why not fit each microtubule separately, calculate its σ, and then average all of the σ values together? In principle, this should return the same σ value as the current method. At the moment, I'm not convinced that individual MTs have tapered ends, primarily because I have not been shown data on individual MTs, and I don't understand where the tapering comes from.

Figure 2

3) Figure 2C has some of the strongest data in the paper--it's clear that EB1-GFP is binding preferentially to the "disrupted" microtubules. The analysis method for this data is to divide the EB1-GFP signal by the underlying tubulin signal, with the idea being that tubulin signal will be dimmer in "disrupted" areas. But how does the labeling ratio of the tubulin affect this analysis? Due to the Poisson statistics of labeling and label incorporation, there are always dimmer areas on labeled microtubules. If the label reduces the affinity of EB1-GFP for microtubules at all, you'll see increased EB1-GFP binding in dim areas, which is perhaps why there is an upward spread in the control data. In any case, how is the uneven fluorescence of the microtubules accounted for in their analysis?

Figure 3

4) I'm confused by their computational model. We know the EB binding site is the vertex of 4 tubulin dimers, but the model seems to allow binding to incomplete sites with fewer than 4 dimers. In Figure 3F, it appears that a single tubulin dimer can bind to an EB protein. In other words, the model does not distinguish between complete and incomplete binding sites. I don't understand how this is realistic. The model measures the distance between EB and its site and scores a hit when the distance is only 1 nm, regardless of whether the site is complete. But presumably the affinity of EB1 for its site will fall if the site is incomplete - there are contact surfaces missing. When you are down to a single tubulin, the affinity may be so low that the interaction is equivalent to a transient, non-specific interaction.

5) I don't know what a normalized on-rate is. What are the units? I'm used to seeing on-rates with units of events per unit concentration per unit time. What is the diffusion limit and how was it calculated? Is it the Smoluchowski limit?

[Editors’ note: what now follows is the decision letter after the authors submitted for further consideration.]

Thank you for submitting your article "Structural state recognition facilitates tip tracking of EB1 at growing microtubule ends in cells" for consideration by *eLife*. Your article has been reviewed by Anna Akhmanova as the Senior Editor, a Reviewing Editor, and two reviewers. The following individuals involved in review of your submission have agreed to reveal their identity: Manuel Thery (Reviewer #2).

The reviewers have discussed the reviews with one another and the Reviewing Editor has drafted this decision to help you prepare a revised submission.

Summary:

The authors have resubmitted their manuscript on the binding of EB1-GFP to microtubules and have included substantial new data.

The resubmission does a good job of clarifying the authors' arguments and of distinguishing between "arrival events" and "binding." Overall, after reading the revised manuscript, we came away thinking that the authors might be on to something. This data should be published so that the field can grapple with it and what it might mean.

That said, the reviewers and myself have concerns about some of the new data, in particular the binding experiments of EB1-GFP to different microtubule lattices. Also, in its current form the manuscript makes several bold claims that are not well supported and does not adequately discuss the structural biology that is relevant to the topic. These problems should be straightforward to fix.

Essential revisions:

1) GDP-Pi microtubules.

We were not convinced that you are measuring the change in affinity due to changes in the lattice by comparing GDP microtubules in the presence of KPO4 vs KCl. The ionic strength of these two salts is different and from your methods you appear to have no buffer (e.g. Tris, Hepes etc) in your imaging conditions. We would expect small changes in ionic strength or pH to have large effects on EB1 binding. We also note than many in the field use acetate rather the chloride ions.

We think you need to go all-in on these nucleotide experiments or remove them from the manuscript. All-in means:

- careful controls for the buffers (e.g. matched ionic strengths, pHs)

- use phosphate analogs (BeFx etc) so that binding can be done under similar buffer conditions

- titration of EB1 to determine affinities

Essentially, experiments to the standard of Maurer 2011 or similar.

2) Positive control data. In previous reviews we asked for positive control data on EB1-GFP. The manuscript now includes data showing that their EB1-GFP can track growing ends (previously not included), but true positive control data is still missing, e.g., the comet decay length as a function of microtubule growth rate.

3) Structural biology. There is no discussion of expansion, compaction, lattice twist, etc. The Zhang et al., 2015 paper is cited, but only to support their modeling of the EB binding site and, oddly, to support the claim than microtubules have 13 protofilaments (the correct one there is Tilney, 1973). The central idea of Zhang et al., 2015 is that EB recognizes a "compacted and twisted" state. There is also the Manka and Moores, 2018 paper that claimed to capture the GDP-Pi state. Also missing is the McIntosh et al., 2018 paper that provided tomograms of microtubule ends in cells and in vitro. We understand that a lot of this science is not settled and are personally confused about expansion and compaction (e.g., there are conflicting reports about kinesin's in this regard, and the yeast tubulins are oddly behaved). But unsettled debates shouldn't just be ignored.

4) Vinblastine data. The authors add vinblastine to cells and claim that it "blunts" the microtubule ends, which is why they see reduced EB binding. But vinblastine also induces outward curvature of protofilaments. If the protofilaments are separated from each other and outwardly curved, could that be the reason that EB has reduced affinity for them? See also the McIntosh data, in which the microtubules in cells are already relatively blunt. Could you realistically get the magnitude of signal reduction that you observe with only a small increase in relative bluntness? The vinblastine part of the paper took an overly-simplified view of the drug's mechanism of action and did not engage with the structural biology here either.

5) On a related note you include a statement in the abstract "Correspondingly, in cells, the conversion of growing microtubule ends from a tapered into a blunt configuration resulted in reduced EB1 targeting.". We feel this is really FAR too strong compared to the data you have – it is not even clear that there are tapered ends in cells at all (no, if you believe the McIntosh paper), so you need to revise this statement.

6) Putting the biochemistry all together. It seems that the authors have on-rate constants (directly measured + modeled), as well as off-rate constants (from lifetimes in their single molecule assays on various lattices), which yields K_D values. They also have equilibrium data on tips, closed lattices, disrupted lattices, etc., at specific EB concentrations. Do these numbers all line up?

7) Killer experiment. From reviewer #1

I don't know if it's fair to ask, but there is an obvious killer experiment. The author's argument is that EB1 has a higher on-rate to incomplete sites + a higher off-rate. But because the on-rate is proportionally higher than the off-rate, they get an increase in binding at equilibrium (Figure 2C). This argument implies that EB will flux more through the incomplete sites, even if the steady-state occupancy is higher. So, they should be able to test that directly with FRAP. E.g., take the disrupted vs. closed lattices, FRAP them, and observe faster recovery to a higher equilibrium fluorescence.

---

## [Author Response]

[Editors’ note: the author responses to the first round of peer review follow.]

Our decision has been reached after consultation between the reviewers. Based on these discussions and the individual reviews below, we regret to inform you that your work will not be considered further for publication in eLife.A major point of interest in the manuscript is the role of specific microtubule lattice structures (defects/edges) in the initial binding of EB1 to microtubules. One reviewer considered that this, and the implication that nucleotide state of tubulin is not the key feature for recognition, was intrinsically interesting. The other two reviewers felt that you were discussing a more nuanced scenario where partial/edge binding sites provide the first binding interaction for EB1 which is then locked in by arrival of new tubulins. These two reviewers were, however, not convinced that the data in the paper were sufficient to support this model over other possibilities.We discussed the paper among the reviewers at some length and I am afraid the consensus was that it is not possible to ask for revisions at this stage. I include all the reviewers comments below so you can get an idea of whether you want to try a new submission to eLife or to modify the manuscript to submit elsewhere.Reviewer #1:In this manuscript the authors argue that EB1 recognizes specific microtubule lattice structures (presenting the lateral side of tubulin dimers) rather than the nucleotide state of the tubulin (ie GTP versus GDP). Thereby they contradict a long-standing belief in the field. As such this paper is ground breaking and deserves publication in a top quality journal such as eLife.The main arguments allowing them to reach these conclusions are:- EB1 binds more frequently to the plus tips of GMPCPP microtubules (ie microtubules made of GTP-like tubulin) than along the lattice. So the tubulin state is not sufficient to determine the affinity of EB1.- EB1 binds the transition between microtubule seeds and the GMPCPP lattice even more than plus tip. So it is not the tip per se but some structural features that EB1 binds to.- EB1 binds disrupted lattice more than intact lattice. They used EB1-beads and cryo EM to correlate lattice structure and EB1 localization.- EB1 binds the edge of sheet-like microtubule structures rather than the central part of the sheet. Arguing again that it recognizes a structure more than the nucleotide state.- key control experiment: Kif5b, which is supposed to recognize the outer rather than the lateral part of tubulins (is this really clear? Demonstrated previously?), does not bind disrupted lattice more than intact lattice.My concerns are the following:- differences between intact lattice and tips on GDP microtubules are significant but really small. Differences are huge when the lattice is so much damaged that is incompatible with actual microtubules (they would depolymerise immediately). Have these difference any physiological relevance?

In the paper, we used lattice damage as a tool to explore potential differences in EB1 affinity for microtubules based on structure, and did not intend to imply that this level of damage would be typical of microtubules in cells. We have clarified this reasoning in the text, as follows: “These “control” and “disrupted‐structure” populations were described for the three most commonly used in‐vitro microtubule nucleotides: GDP, GMPCPP, and GTPγS. We used the “disrupted‐structure” populations with disruptions in lattice integrity as a tool to reveal potential differences in EB1 binding based on microtubule structure.”

Through our experiments and simulations, we find that the on‐rate of EB1 to edge sites is predicted to be ~70‐fold faster than to lattice sites, which could have a physiological impact at the tips of GTP microtubules, which have been previously shown to be enriched in edge sites.

- Tricks to disrupt the lattice are quite strong and may have unexpected side-effects.

While we do use “disrupted‐structure” microtubule populations to explore the effect of structure on EB1 affinity for the microtubule, the methods used to create these populations reflect commonly used methods to generate in‐vitro microtubules, and so we do not predict that unexpected side‐effects are likely. In addition, the methods used for the three different nucleotides are distinct, and so it is unlikely that similar side effects would occur in each case. We will clarify the text, as follows: “We note that, while the preparation protocols for microtubules using the three different nucleotides were distinct, all of these protocols reflect commonly used methods for producing stable in vitro microtubules.”

- On dynamic GDP/GTP microtubules there is a high concentration of EB1 at the growing tips. Does the nucleotide state also matters? Or only the structure? Does the normalization in Figure 2C (between normal and disrupted lattices) hide the fact that EB1 binds much more to GMPCPP and GTPgammaS microtubules than to GDP microtubule and thus that the tubulin state actually matters "more" than the lattice structure? It would be nice to show the non-normalized values and comment on this point.

We predict that, based on previously published work, EB1 likely has differential affinity to different GTP nucleotide states, and we did not intend to contradict or weigh in on this argument. Because the microtubules in Figure 2C are generated with distinct protocols for each nucleotide type, we believe that direct comparisons of EB1 binding to the “control” populations between nucleotide states would be an over‐interpretation of our data. Therefore, although normalization to the average binding intensity of each respective control seems appropriate for Figure 2C, we will provide the baseline average control binding intensity for each nucleotide in the main text, as follows: “The EB1‐GFP/tubulin ratios for each microtubule were normalized to the grand average EB1‐GFP/tubulin ratio for the control microtubule population in each case, to allow for direct comparisons between the disrupted‐structure data and their respective controls (average EB1‐GFP/tubulin binding values used for normalization were as follows: GMPCPP: 0.098 ± 0.0009; GDP: 0.100 ± 0.006; GTPγS: 0.68 ± 0.007 (mean ± SEM); see Materials and methods section).”

The fact that Kif5b binds the outer part of tubulin is known (at least it is assumed by the authors but I am not sure how this is actually demonstrated).

Electron microscopy work has been completed for the specific monomeric Kif5b construct used in our analysis to demonstrate binding to the outer face of a single tubulin subunit. The following text is included to clarify this point: “… we used a monomeric Kif5B‐GFP kinesin construct…to allow for microtubule binding analysis of the monomeric kinesin (K339‐GFP) (Case et al., 2000; Tomishige and Vale, 2000). The Kinesin‐1 Kif5B protein has been shown to dock with a stoichiometry of one kinesin motor per tubulin heterodimer, and binds to the outer surface of the α or β tubulin monomers, rather than the lateral surfaces, in electron microscopy experiments (Hirose and Amos, 1999; Hoenger et al., 1998; Kikkawa et al., 2000; Kozielski et al., 1998; Löwe et al., 2001; Marx et al., 2006; Moores et al., 2002; Nogales et al., 1999; Rice et al., 1999; Sosa et al., 1997)”

- The fact that EB1 binds more to tappered end than to the lattice is known (Guesdon et al., 2016). The fact that EB1 binds to the side of tubulin dimers is known (Zhang et al., 2015). The fact that Kif5b binds the outer part of tubulin is known (at least it is assumed by the authors but I am not sure how this is actually demonstrated). So, it is no surprise that EB1 binding is structure-sensitive and that Kif5b is not. One could consider that all evidences were already present and thus contest the actual innovation of this work.

We agree that evidence exists in the literature to support our argument that EB1 binding is structure sensitive, which we feel strengthens our conclusions.

- I can't prevent myself from being frustrated that our work, which showed for the very first time some tubulin dynamics on supposedly damaged lattice regions is not cited here. I hope this is not intentional to claim for more novelty.

We apologize for this inadvertent omission, and this work is now cited in the introduction, as follows: “… recent work has discovered that lattice damage and tubulin turnover can occur on the microtubule lattice, leading to irregularities along the lattice itself (Schaedel et al., (2015)).”

- The authors did not discuss the work of Pous and Perez, 2016 in which they showed that Clip170 binds to damaged microtubule lattice. We should also take into account that the observations in Pous and Perez paper somehow limit the novelty of this work.

We have added the following description of this work to the discussion: “…it has been shown that microtubule lattice defects resulting from microtubule‐microtubule contacts may be recognized by CLIP‐170 to stimulate microtubule rescue (DeForges, 2016).”

However, we feel that the primary novelty of our new work is in dissecting how the binding location of EB1 on the microtubule may lead to structural recognition based on a steric‐hindrance‐mediated onrate. This has now been clarified throughout the text and in the paper title.

Reviewer #2:The submission from Reid et al., addresses an interesting question about how EB1 recognizes the growing microtubule end. Although there has been a lot of work on this question from multiple groups, I think it is fair to say that the science is not exactly settled. In their work, the authors use quantitative fluorescence microscopy along with computational simulations of association and methods developed by the Gardner lab to create microtubules with 'damaged' regions (disruptions in the lattice). The logic of the manuscript goes as follows. Tow experimental observations (i) that EB1 prefers to bind more tapered microtubule ends, which correlates with the plus end; EB1 also localizes preferentially at the transition between seed and new growth, which may contain lattice defects (Figure 1); and (ii) that EB1 prefers to bind microtubules containing disruptions (Figure 2); motivate the hypothesis that EB1 may prefer to associate with 'edge' sites. An implementation of this hypothesis into a computational simulation of association kinetics shows that edge association can lead to faster association (Figure 3), and control simulations using a 'face binder' (and a control experiment using kinesin) demonstrate that edge binding is not an obligatory outcome of this kind of modeling. This leads to a model (Figure 5) wherein EB1 arrives via a 'lock-in' mechanism that entails rapid (but weak/transient) associations with partial/edge-binding sites that can get 'locked in' to a higher affinity state by arrival of new ab-tubulins that complete the EB1 binding site.I appreciate that the authors are looking at this question in a new way, and deploying new approaches to do it. The manuscript is well-written, and the figures are logical and clear and obviously prepared with care. However, there appear to be some inconsistencies in logic/interpretation that make me question whether all of the data add up in the way the authors say they do.

The proposed responses to these issues are addressed in detail below.

Additionally, and independent of my concern about inconsistency, given that this is a pretty well-studied problem already, I don't think that in its present form the manuscript has done enough to convincingly rule out alternative mechanisms (e.g. that EB simply associates directly with 'cap' sites without going through a 'lock-in'), to justify the need for the alternative model presented, or to show it can quantitatively explain existing data from other groups (e.g. Maurer et al., 2014).

Once we received the reviews for our manuscript, we realized that we did not properly convey our perspective regarding the contribution of our work to the discussion regarding EB1 tip tracking. In general, we did not intend to put forward a new model to overturn existing models for EB1 tip tracking. Essentially, we feel that the idea that the on‐rate of EB1 to microtubules may be sensitive to microtubule structure is interesting, and could potentially speed the arrival of EB1 to microtubule plusends. In providing the final (speculative) cartoon, our attempt was to integrate this general idea with current models for EB1 tip tracking. However, the final cartoon was perhaps overly suggestive that we were proposing an alternate model for tip tracking. Thus, we plan to drop the final cartoon, and to limit our discussion of “tip tracking” to a short paragraph in the Discussion section. Further, we will refocus the manuscript on our key result, which is that the on‐rate of EB1 to microtubules may be sensitive to the structure of the microtubule itself.

In the end, while it may be that EB1 can rapidly and transiently associate with edge sites, I'm not yet convinced that is necessary for the mechanism of its end binding.

We agree that data was not presented in the manuscript to suggest this mechanism is necessary for tip tracking of EB1. As noted above, we have refocused the manuscript on our key result regarding the onrates of EB1 to microtubules based on structure. To specifically address this concern, we will also add the following sentence to the discussion: “Overall, while we feel that this process will contribute to rapid binding of EB1 to the microtubule tip, starting at the open, tapered growing end with its more easily accessible edge sites, its specific contribution to the mechanism of tip tracking remains an open question.”

Essential revisions:Connection between experiments and simulations.The experiments showing association of EB1 with damaged lattices are interesting. But the molecular nature of those sites is not known. The authors pursue a model in which the defining characteristic of those sites is elevated edge content. But can they rule out that in those damaged sites tubulin adopts more 'sheet-like' conformations that mimic end taper and curvature (e.g. Guesdon)?

Based on our analysis, we predict that the damaged lattices mimic end‐taper and “sheet‐like” conformations, which would correspondingly increase the number of edge sites, and we did not observe excessive curvature in our electron microscopy analysis of the damaged lattices. However, we will add the following sentence to the discussion to address this concern “… we cannot exclude that curvature of microtubule sheets could potentially act to alter EB1 on‐rates.”

- They simulate association kinetics, but nearly all experiments measure binding in a way that probably reflects affinityThe simulations (for which the use of blunt and tapered microtubule segments is clever) simulate the rate of association in what I think is a reasonable way. Mostly they are clear about this, but in a few places the work 'binding' is used in a way that could be misleading.

We agree that the simulation itself does not measure affinity or steady‐state binding, and that the term “binding” in regards to the simulation could be misleading. We have now renamed simulated EB1 “binding” as simulated EB1 “arrival events” throughout the manuscript text and in the figures, to clarify that the simulation does not reflect steady‐state binding, but rather the locations where EB1 arrives (in proper binding orientation and position) most rapidly onto the microtubule.

The authors reasonably state that these partial edge binding sites are likely to be low-affinity ones, but multiple supporting figure panels essentially reflect equilibrium binding, which should show the high-affinity sites, not the low-affinity ones the authors care about. This apparent contradiction needs to be clarified.

This is a good point, and has caused us to think carefully about our results. We propose adding the following comment to the discussion: “After binding to an edge‐site, EB1 would likely undergo a higher off‐rate as a result of the partial binding interface that is inherent to these edge‐sites. However, our results show an increase in steady‐state binding to disrupted microtubule structures, regardless of nucleotide state (Figure 2C‐E), suggesting that the affinity of EB1 is higher on disrupted structures, regardless of the likely increase in off‐rates. This increase in affinity may be due to a stronger effect of disrupted microtubule structure in accelerating the EB1 on‐rate, as compared to increasing the EB1 offrates. For example, we observed a ~45‐fold increase in EB1 on‐rates to disrupted GDP microtubule structures relative to controls (Figure 3J), while the “short” EB1 dwell times, which could be associated with edge‐bound EB1 on GDP microtubules, were only ~8 fold faster than the lattice‐bound “long” dwell times (Figure S2D), resulting in a net affinity increase of ~5‐fold.”

I think the details of the 'lock-in' also need to be fleshed out in terms of the relative kinetics of EB1 dissociation from edge sites and ab-tubulin association to complete EB1 binding sites. Under what regimes can this work?

We agree that this is an interesting question, however, the “lock‐in” model was only speculative at this point, and further experimental work will be required to determine whether a “lock‐in” model contributes to EB1 tip tracking of growing microtubule plus‐ends. To focus the manuscript on our current data and key results, we have dropped the final cartoon (and associated text) that included this speculative idea.

- Integration with current knowledgeMaurer et al., 2014 contains a minimal kinetic model that can describe quantitative observations about EB1 comets. The authors are essentially proposing a different model for how EB1 gets to its comet site. I think they need to explain better why an alternate model is needed, and maybe even show that theirs can actually do what Surrey's toy model did.

As noted above, we did not intend to discard existing models for the general mechanism of EB1 tip tracking, but rather to describe how the arrival rates of EB1 to microtubules may be influenced by microtubule structure. We have refocused the manuscript on our key result regarding the on‐rates of EB1 to microtubules based on structure, which is supported both experimentally and in simulation. In addition, we have added the following comment to the discussion: “Overall, while we feel that this process will contribute to rapid binding of EB1 to the microtubule tip, starting at the open, tapered growing end with its more easily accessible edge sites, its specific contribution to the mechanism of tip tracking remains an open question.”

In subsection “Rapid frame-rate fluorescence microscopy experiments demonstrate increased on-rate of EB1-GFP to disrupted-structure microtubule pools”, the wording seems to indicate that Guesdon et al., also observed significant edge binding. But if I read Guesdon et al., correctly, only 2.1% of their observed binding was at an edge or on the inside of a sheet. This is a borderline minor point, but I think they authors need to be clearer about where prior data support their model, and about where their data appear to be in conflict with prior data.

There are important differences between the observed binding percentages between our new work and that of Guesdon et al., which are: (1) we have normalized the binding percentages to the number of available sites in each case. For example, the number of full lattice sites far exceeds the number of edge sites, and so by normalizing to the number of available sites in each case, our normalized binding frequencies would be substantially larger than the absolute binding frequencies as reported by Guesdon et al., and (2) the work of Guesdon et al., was performed on dynamic microtubules, rather than stabilized microtubules as in our work, and so while new edges would continually be generated during growth, the lifetime of individual edge sites would likely be much more transient than that of full lattice sites. Therefore, for a direct percent‐binding comparison to the stabilized microtubules in our work (in which the time availability of each site is equal), the percentages in Guesdon et a.,l would need to be normalized to account for the total time availability of each site. Thus, a direct comparison of binding percentages between our work and that of Guesdon et al., would require further analysis. However, we have clarified the wording regarding this reference, as follows: “Qualitatively, we observed EB1‐beads that bound to microtubule sheets, both in the middle of the sheet, and on the edges of sheets (Figure 3I, left), as has been previously observed on dynamic microtubules (Guesdon et al., 2016).

Reviewer #3:Reid et al. have studied how EB1-GFP binds to different microtubule structures using TIRF microscopy, electron microscopy, and computational modeling. They show that EB1-GFP binds to the ends of GMPCPP microtubules at intermediate salt concentrations and binds preferentially to "disrupted" microtubules that were prepared with Taxol. The idea is that these structures have incomplete EB binding sites, which is why EB prefers them. Their computational model is a Brownian dynamics simulation that measures the amount of time that EB diffuses become coming close to its binding site. They propose that EB binds with faster on-rates to incomplete binding sites located at the very end of the microtubule. They conclude with a model for EB end-recognition wherein the tapered, incomplete end of the lattice is the location where EB loads onto the microtubule.It's important to state that this manuscript is squarely at odds with some ideas circulating in the field. A few labs (e.g, Surrey) have published data showing that the EB signal is displaced back from the very end of the microtubule, with polymerases like XMAP215 being found at the very end instead. There are a few hypotheses for why EB is displaced. One hypothesis is that EB prefers GDP-Pi lattices (Nogales). My favorite idea is that the end is tapered, which means that the binding sites for EB1 are incomplete, and thus EB doesn't bind there. I should say that a recent paper by Guesdon et al., used EM to show that EB can bind at the very end, but the fluorescence data is very hard to ignore, given the much larger number of observations relative to the EM. This manuscript argues that the "EB-free zone" is not EB free and is rather the zone that EB uses to load onto the lattice.

While we have refocused the manuscript text to emphasize our primary conclusion regarding the ability of EB1 to detect microtubule structure, we do not feel that a higher on‐rate of EB1 onto edge sites at microtubule tips would necessarily be at odds with the previously described peak concentration of EB1 away from the very end of the microtubule, due to the difference in binding site density for edge sites as compared to full lattice sites. We have added the following explanation regarding this issue to the Discussion section:

“Overall, we propose that [structural recognition by EB1] will promote rapid loading of EB1 at the microtubule tip, starting at the open, tapered growing end with its more easily accessible edge sites. Although new edge sites are continuously generated as a result of tubulin subunit additions to growing microtubule plus‐ends, we note that the density of edge sites on an open, tapered microtubule lattice would be ~7‐fold lower than that of lattice sites for a closed microtubule tube (ratio 2:14 edge sites: total binding sites per tubulin layer), and so regardless of whether structural recognition facilitates rapid loading of EB1 onto edge sites at the microtubule tip, the peak in EB1‐GFP intensity for a plus‐end EB1GFP comet may be penultimate to the tapered tip (as previously described, Maurer et al., 2014), in a location where a high density of EB1 molecules are bound to the much higher density, four‐GTP‐tubulin pocket lattice binding sites, potentially via addition of new, incoming GTP‐tubulin subunits onto the location of previously edge‐bound EB1 molecules.”

Overall, the manuscript does a poor job engaging with the debate. Also, in order to overturn an emerging consensus, the expectation re: data and analysis is high. Unfortunately, I'm not sure that the paper brings enough to the table.

We have now clarified the Abstract, Results section, and Discussion section to focus on our primary result, which is that structural recognition substantially increases the on‐rate of EB1 to microtubules. While we feel that this is an exciting and important new concept, we do not present evidence to suggest that current EB1 tip tracking models, such as the increased affinity of EB1 to GTP‐tubulin as opposed to GDP‐tubulin, should be overturned or discarded. In addition, we did not intend to suggest that this mechanism could exclusively explain tip‐tracking of EB1, but rather that structural recognition could help to facilitate rapid arrivals of EB1 to the microtubule plus‐end. We apologize for this lack of clarity, and, in addition to dropping the final speculative cartoon, we have added the following sentence to clarify our thinking in this regard: “Overall, while we feel that this process will contribute to rapid binding of EB1 at the microtubule tip, starting at the open, tapered growing end with its more easily accessible edge sites, its specific contribution to the mechanism of tip tracking remains an open question.”

I will highlight some technical concerns that I have below. But I'm surprised to have a paper on EB binding that doesn't have any analysis of end-tracking of dynamic microtubules in it. There is no direct evidence that the tapered end is where the EB loads on, which could only come from clever experiments using dynamic microtubules. Combined with the problems highlighted below, I cannot support publication in eLife.

As noted above, we have refocused the manuscript to emphasize the primary result, which is that the nature of EB1’s binding site on the microtubule allows for microtubule structure recognition as a result of steric hindrance in binding to its four‐tubulin pocket binding site in the lattice. However, we agree that the inclusion of dynamic microtubule results would strengthen the manuscript. Therefore, we have completed dynamic microtubule experiments in which we compared average EB1‐GFP binding intensities at growing microtubule plus‐ends, on the GDP lattice, on the GMPCPP‐seed, and at the transition between the GDP‐lattice and the GMPCPP‐seed. We found that, as would be expected, EB1GFP intensity was ~2.3‐fold (230%) higher at growing microtubule plus‐ends as compared to the overall average intensity on the GDP lattice and GMPCPP seeds. However, consistent with our conclusion that EB1 may recognize microtubule structure, the EB1‐GFP fluorescence at the seed/GDP‐lattice transition point was 38% higher than the average intensity on the GDP lattice and GMPCPP seeds (p<10^‐5^ vs GDP lattice, p<10^‐6^vs GMPCPP seeds, n=134 microtubules). These results are consistent with our conclusion that EB1 may recognize microtubule structure, even in the presence of free tubulin in a dynamic microtubule assay. However, EB1‐GFP binding at growing microtubule plus‐ends remained substantially higher than transition point binding (p<10^‐16^), allowing us to reiterate that this finding may contribute to, but does not overturn, current models for EB1 tip tracking.

Figure 11) I would like to see some positive control data on their EB1-GFP. Does the protein form nice comets that match the literature in terms of size, etc? Does the salt concentration they use affect end-tracking behavior? Are the EB1-GFP signals in Figure 1B and Figure 1F single molecules? They look like larger puncta. If they are not single molecules, are the larger signals forming by accumulation or are they aggregates? Quality control of this type would be very welcome.

Dynamic microtubule experiments have been added at two different salt concentrations encompassing experimental work in the manuscript, both of which exhibited robust end tracking behavior, with no evidence of EB1 protein aggregates. The Figure 1 data was not intended as single‐molecule data, but we see no evidence of aggregation or aggregates in our new time‐lapse data. We have now also included images over a range of EB1‐GFP signal intensities in Figure 1, to demonstrate a robust result regardless of EB1‐GFP intensity.

2) I was surprised by the observation of extensive tapering on the ends of the GMPCPP microtubules in Figure 1C. Where does this tapering coming from? The degree of tapering observed by TIRF microscopy (on the order of 1.5 um) is much larger than the degree of tapering observed by EM, where the length of the taper is similar to the width of the MT (on the order of 25 nm). Why does the degree of tapering differ so much? Microtubule ends are sometimes ragged after they are broken during pipetting. But even for GMPCPP microtubules, these ragged ends shrink back to blunt ends within a few minutes. So my concern is that the tapering in Figure 1C is coming from the computational procedure wherein the microtubules were aligned and then rebinned to a common length. If the end of each microtubule is not at the same location within the bins, this will create a spreading out of the fluorescence signal, won't it? The methods did not provide enough information about the alignment and binning procedure for me to rule this out. The methods did make clear that the fitting of the data to the error function happened only after the alignment and rebinning. Individual microtubules were not fit to the error function, which means that the alignment procedure did not have sub-pixel accuracy. Why not fit each microtubule separately, calculate its σ, and then average all of the σ values together? In principle, this should return the same σ value as the current method. At the moment, I'm not convinced that individual MTs have tapered ends, primarily because I have not been shown data on individual MTs, and I don't understand where the tapering comes from.

We thank the reviewer for this observation – yes, the calculated tip standard deviations were large, and although the microtubule fluorescence binning procedure (which has now been more clearly described in the methods) equally redistributes microtubule fluorescence into a standard number of bins regardless of microtubule length, we were performing the analysis over a very large range of microtubule lengths, which artificially inflated the calculated tip standard deviation values. While fitting to individual microtubules was too noisy due to stochastic labeling, we addressed this issue by performing the analysis on small groups of similar length microtubules (maximum microtubule length standard deviation = 0.7 μm in each case). In addition, we verified that our calculated tip standard deviations were statistically indistinguishable even if a group of microtubules with a very small microtubule length standard deviation was used (maximum microtubule length standard deviation = 0.2 μm). While the result that tip standard deviations were larger on the microtubule ends with brighter EB1‐GFP signals remained unchanged with our new analysis, the calculated tip standard deviations are now substantially smaller, in line with our EM observations in Figure 1E.

Figure 23) Figure 2C has some of the strongest data in the paper--it's clear that EB1-GFP is binding preferentially to the "disrupted" microtubules. The analysis method for this data is to divide the EB1-GFP signal by the underlying tubulin signal, with the idea being that tubulin signal will be dimmer in "disrupted" areas. But how does the labeling ratio of the tubulin affect this analysis? Due to the Poisson statistics of labeling and label incorporation, there are always dimmer areas on labeled microtubules. If the label reduces the affinity of EB1-GFP for microtubules at all, you'll see increased EB1-GFP binding in dim areas, which is perhaps why there is an upward spread in the control data. In any case, how is the uneven fluorescence of the microtubules accounted for in their analysis?

We have now clarified the description of the data that makes up Figure 2 to note that we did not perform a spatial analysis of EB1 distribution along the length of each microtubules, as follows: “We…measured the average EB1‐GFP fluorescence intensity over background for each microtubule, which was normalized to the average Rhodamine‐tubulin fluorescence intensity over background for each microtubule…. The average EB1‐GFP/tubulin ratios for each microtubule were then normalized to the grand‐average EB1‐GFP/tubulin ratio for the control microtubule population in each case, to allow for comparisons between the disrupted‐structure data and the controls.” We also note that the degree of labeling was consistent between the controls and disrupted‐structure microtubules (microtubules were made from a common reaction mixture for each control/disrupted experiment), and so if the rhodamine label reduced the affinity of EB1‐GFP for microtubules, this effect would be similar in both cases.

Figure 34) I'm confused by their computational model. We know the EB binding site is the vertex of 4 tubulin dimers, but the model seems to allow binding to incomplete sites with fewer than 4 dimers. In Figure 3F, it appears that a single tubulin dimer can bind to an EB protein. In other words, the model does not distinguish between complete and incomplete binding sites. I don't understand how this is realistic.

As noted above, the simulation does not measure affinity or steady‐state binding rates, so the term “binding” could be misleading. We have now renamed simulated EB1 “binding” as EB1 “arrival events” throughout the manuscript text and in the figures, to clarify that the simulation does not reflect steadystate binding, but rather the locations in which EB1 arrives (in proper binding position and orientation) most rapidly onto the lattice. In Figure 3F, we explore the number of arrival events of EB1 to each microtubule type (control/disrupted) per every 4000 simulated EB1 molecules, regardless of the tubulin binding site configuration (1‐4 tubulin subunits). Thus, we have clarified the y‐axis in panel D – now called “total arrival events per microtubule”. The purpose of this analysis was to determine whether there would be a differential arrival rate of EB1 to different binding site configurations, regardless of the subsequent off‐rate or dwell time of EB1 at that particular site.

The model measures the distance between EB and its site and scores a hit when the distance is only 1 nm, regardless of whether the site is complete. But presumably the affinity of EB1 for its site will fall if the site is incomplete - there are contact surfaces missing. When you are down to a single tubulin, the affinity may be so low that the interaction is equivalent to a transient, non-specific interaction.

As noted above, we have clarified our wording to note that the computational model does not account for off‐rates and/or affinity of a particular site, but rather, the model specifically explores whether the on‐rate of EB1 to microtubules may be affected by microtubule structure. However, we note that in panels 3G and 3H, we present a specific comparison of simulated EB1 arrivals to 2‐tubulin vs 4‐tubulin sites, excluding 1‐tubulin sites that could potentially have a high off‐rate, and thus limiting our analysis to the partial binding sites that have been observed using electron microscopy in our work as well as in previous work (Guesdon et al.,). Here, we find that there is a ~70‐fold increase in the arrival rate (in proper orientation and position) to 2‐tubulin sites as compared to 4‐tubulin sites.

5) I don't know what a normalized on-rate is. What are the units? I'm used to seeing on-rates with units of events per unit concentration per unit time.

We agree that term “normalized on‐rate” in this graph was too vague. This term has now been dropped, and the units in panel 3H are now shown as “Fraction of Simulated EB1 Molecules”. In this simulation, the ultimate fate of each of the 200,000 simulated EB1 molecule was recorded (e.g., an EB1 molecule never arrives to a microtubule, an EB1 molecule arrives at 2‐tubulin site, an EB1 molecule arrives at a 4tubulin site, etc). Therefore, the numbers in panel 3H show a direct comparison of the fraction of simulated EB1 molecules that arrived at a 2‐tubulin (edge) site, as compared to the fraction that arrived at a 4‐tubulin (lattice pocket) site in the simulation. While these numbers reflect the relative on‐rates, they are not measured in terms of EB1 concentration or arrivals per unit time, and so the term “on‐rate” has been dropped.

What is the diffusion limit and how was it calculated? Is it the Smoluchowski limit?

The term “diffusion limit” has now also been altered, to avoid confusion. The intent with the “diffusion limit” value was to demonstrate that the 2‐tubulin (edge) site EB1 arrivals were still substantially (~4fold) less frequent than the 1‐tubulin site EB1 arrivals. The 1‐tubulin site arrivals are suggestive of a “diffusion limit” in that the steric hindrance to a 1‐tubulin site arrival would be negligible, however, as noted above, arrival to these sites was still stereospecific, such that EB1 could not bind “upside down”, even to a 1‐tubulin site. This comparison is now clarified in the text.

[Editors’ note: the author response to the re-review follow.]

Thank you for submitting your article "Structural state recognition facilitates tip tracking of EB1 at growing microtubule ends in cells" for consideration by eLife. Your article has been reviewed by Anna Akhmanova as the Senior Editor, a Reviewing Editor, and two reviewers. The following individuals involved in review of your submission have agreed to reveal their identity: Manuel Thery (Reviewer #2).The reviewers have discussed the reviews with one another and the Reviewing Editor has drafted this decision to help you prepare a revised submission.Summary:The authors have resubmitted their manuscript on the binding of EB1-GFP to microtubules and have included substantial new data.The resubmission does a good job of clarifying the authors' arguments and of distinguishing between "arrival events" and "binding." Overall, after reading the revised manuscript, we came away thinking that the authors might be on to something. This data should be published so that the field can grapple with it and what it might mean.That said, the reviewers and myself have concerns about some of the new data, in particular the binding experiments of EB1-GFP to different microtubule lattices. Also, in its current form the manuscript makes several bold claims that are not well supported and does not adequately discuss the structural biology that is relevant to the topic. These problems should be straightforward to fix.Essential revisions:1) GDP-Pi microtubules.We were not convinced that you are measuring the change in affinity due to changes in the lattice by comparing GDP microtubules in the presence of KPO4 vs KCl. The ionic strength of these two salts is different and from your methods you appear to have no buffer (e.g. Tris, Hepes etc) in your imaging conditions. We would expect small changes in ionic strength or pH to have large effects on EB1 binding. We also note than many in the field use acetate rather the chloride ions.We think you need to go all-in on these nucleotide experiments or remove them from the manuscript. All-in means:- careful controls for the buffers (e.g. matched ionic strengths, pHs)- use phosphate analogs (BeFx etc) so that binding can be done under similar buffer conditions- titration of EB1 to determine affinitiesEssentially, experiments to the standard of Maurer 2011 or similar.

We thank the reviewers for noting this concern, which we had not previously considered. In order to perform experiments with matched ionic strengths, with have now added new data with a phosphate analog, which, as the reviewer notes, has now allowed us to perform detailed binding experiments at matching ionic strengths and salt molarities. The new figures and manuscript text have been copied below. We note that, because the results with the analog were very similar to our original results, we kept our original data in the manuscript, which had matching salt molarities, even though the ionic strengths were not matched in these experiments. However, we clearly note the concern regarding mismatched ionic strengths in the text, to motivate the use of the phosphate analog for the subsequent data set (manuscript text copied below). In addition, due to the environmental and safety hazards of using Beryllium in the lab, we have selected an alternative analogue, AlF_4_^-^, described alongside BeF_3_^-^ in Carlier et al., 1988.

Subsection “EB1 tip tracking by microtubule structure recognition”, introduction to analogue experiments:

“In the KPO_4_ experiments (Figure 5A-C), while the molarity of the salts in the experiments were matched (55 mM KPO_4_ vs 55 mM KCl), the ionic strengths of the reaction mixtures were mismatched, which could influence EB1 binding to the lattice. Therefore, we performed further experiments using an analogue of phosphate, AlF_4_^-^ (Antonny and Chabre, 1992; Carlier et al., 1988; Petsko, 2000). Here, the molarity and ionic strengths of the salts included in both experiments were identical (52.5 mM KCl and 2 mM NaF for both experiments). However, 10 μM AlCl_3_ was included in the phosphate analogue experiments, allowing the Al^3+^ and F^-^ ions form the π analogue AlF_4_^-^, which mimics a phosphate group, but with 1000-fold higher affinity (Antonny and Chabre, 1992; Carlier et al., 1988; Petsko, 2000).”

Subsection “EB1 tip tracking by microtubule structure recognition”, next paragraph, new results text:

“Similar to the KPO_4_ experiments, we observed reduced EB1 binding in the GDP+AlCl_3_ experiments as compared to the GDP-alone experiments (Figure 5D). We then quantified EB1-GFP binding using our previously described automated analysis tool (Reid et al., 2017) under conditions of increasing EB1 concentrations (Figure 5E). As previously described, the EB1 binding data was best fit using a cooperative binding curve (Lopez and Valentine, 2016; Zhu et al., 2009), and so we fit the results in each case to a Hill equation, which demonstrated a ~2-fold lower affinity of EB1 for the GDP-Pi-Analogue relative to the GDP microtubules (Figure 5E, see Materials and methods; K_d,GDP_=1.9x10^5^ nM, n_GDP_=2.2; K_d,Analogue_=4.2x10^5^.”

2) Positive control data. In previous reviews we asked for positive control data on EB1-GFP. The manuscript now includes data showing that their EB1-GFP can track growing ends (previously not included), but true positive control data is still missing, e.g., the comet decay length as a function of microtubule growth rate.

We have added positive control data, in the form of a comet length vs microtubule growth rate graph, as new Figure 1—figure supplement 1D, as follows:

3) Structural biology. There is no discussion of expansion, compaction, lattice twist, etc. The Zhang, 2015 paper is cited, but only to support their modeling of the EB binding site and, oddly, to support the claim than microtubules have 13 protofilaments (the correct one there is Tilney, 1973).

The reference regarding 13 protofilaments has now been corrected, although Zhang, 2015 is cited at multiple locations in the paper as we used the Cryo-EM structural data from this paper to model the EB1 binding site on the microtubule (Figure 3B, left; source data from Zhang et al., 2015).

The central idea of Zhang, 2015 is that EB recognizes a "compacted and twisted" state. There is also the Manka and Moores, 2018 paper that claimed to capture the GDP-Pi state. Also missing is the McIntosh, 2018 paper that provided tomograms of microtubule ends in cells and in vitro. We understand that a lot of this science is not settled and are personally confused about expansion and compaction (e.g., there are conflicting reports about kinesin's in this regard, and the yeast tubulins are oddly behaved). But unsettled debates shouldn't just be ignored.

We have attempted to address these issues regarding the underlying structural biology, as follows:

a) Reviewer: “The central idea of Zhang et al., 2015 is that EB recognizes a "compacted and twisted" state. There is also the Manka and Moores, 2018 paper that claimed to capture the GDP-Pi state.”

In the previous version of the paper, we mentioned the Manka and Moores, 2018 paper within the Results section as part of our GDP-Pi results. However, we have expanded our comments to include the Zhang et al., 2015 paper. Given that this structural biology connection seemed quite speculative on our part, we have kept our comments quite brief. Following is the manuscript text:

“…These results could perhaps be explained by previous cryo-electron microscopy findings in which it was demonstrated that the GDP-P_i_ microtubule structural state is distinct from that of GDP microtubules (Manka and Moores, 2018b). Further, published cryo-EM structures of microtubules co-assembled with EB3 show that EB1 may promote a compacted microtubule lattice with a unique twist, once it is bound to the microtubule (Zhang et al., 2015). This structural transition could account for the ultimate destruction of the stable 4-tubulin binding site of EB1 as the lattice-bound tubulin subunits transition from GTP to GDP-Pi conformations, perhaps promoted by the binding of EB1 to the microtubule. We conclude that EB1 does not likely directly bind to GDP-P_i_ microtubules, but, instead, that hydrolysis of GTP-tubulin to GDP-P_i_ -tubulin could initiate formation of a low-affinity EB1 binding site.”

b) Reviewer “Also missing is the McIntosh, 2018 paper that provided tomograms of microtubule ends in cells and in vitro.”

We thank the reviewers for noting this omission, and have added the following comment to the Discussion section:

“…a recent electron microscopy study reported that the tips of growing microtubules within mitotic spindles display a gentle outward curvature of individual protofilaments (McIntosh et al., 2018). This outward curvature could enhance a structural recognition mechanism by increasing the availability of exposed protofilament edge sites at growing microtubule ends, due to the increased separation between individual protofilaments.”

4) Vinblastine data. The authors add vinblastine to cells and claim that it "blunts" the microtubule ends, which is why they see reduced EB binding. But vinblastine also induces outward curvature of protofilaments. If the protofilaments are separated from each other and outwardly curved, could that be the reason that EB has reduced affinity for them? See also the McIntosh data, in which the microtubules in cells are already relatively blunt. Could you realistically get the magnitude of signal reduction that you observe with only a small increase in relative bluntness? The vinblastine part of the paper took an overly-simplified view of the drug's mechanism of action and did not engage with the structural biology here either.

In Figure 6(A-C), we quantitatively estimated the microtubule protofilament tip extension in our LLC-Pk1 cells, and found that tip protofilament extensions were decreased with increasing concentrations of Vinblastine (Figure 6A-C). Therefore, at least in interphase LLC-Pk1 cells, it may be that Vinblastine ultimately leads to blunting of the growing microtubule end, rather than stabilization of outwardly curved protofilaments. Further, based on our model, stabilization of separated, outwardly curved protofilaments would be predicted to increase the affinity of EB1 for the microtubule ends by increasing the density of exposed protofilament edge sites, rather than reduce the affinity.

Data from the McIntosh et al., 2018 paper was presented for kinetochore or interpolar microtubules in mitosis, which, due to the physical and regulatory constraints within the mitotic spindle, may not be representative of the rapidly growing interphase microtubules in the LLC-Pk1 cells that were used in our study. Regardless, extended, flared, protofilaments from 30-54 nm in length were observed in the McIntosh study, which, especially given the separation of protofilaments, we would predict may increase targeting of EB1 to growing microtubule plus-ends in untreated cells, based on our current results.

That said, we acknowledge that our cellular results are correlative, and that there could be other effects of the drug that could alter EB1 binding at the growing microtubule plus-ends. Thus, we agree that our conclusions should be softened accordingly. Thus, we have reviewed all text relating to the cell experiments in Figure 6, and updated text wording regarding the conclusions, as follows:

Subsection Title changed to:

“Blunting of tapered microtubule tip structures correlates with suppressed EB1 binding in cells”

Experiment motivation:

“We then asked whether blunting of the microtubule tip structure, leading to a reduced number of protofilament-edge binding sites, would be correlated with the efficiency of EB1 tip tracking.”

Results section conclusion:

“We conclude that the efficiency of cellular EB1-GFP tip tracking is disrupted with Vinblastine treatment, perhaps due to the blunting of tapered tip structures at growing microtubule ends, which cuts the availability of high-accessibility EB1 protofilament-edge binding sites. However, we cannot exclude that other structural or chemical cellular responses to Vinblastine may be at play as well.”

5) On a related note you include a statement in the abstract "Correspondingly, in cells, the conversion of growing microtubule ends from a tapered into a blunt configuration resulted in reduced EB1 targeting.". We feel this is really FAR too strong compared to the data you have – it is not even clear that there are tapered ends in cells at all (no, if you believe the McIntosh paper), so you need to revise this statement.

As noted above, we quantitatively estimated the tip taper in control cells and upon Vinblastine treatment of these cells (Figure 6A-C). Regardless, we agree that the Abstract statement was too strong. Thus, we have revised as follows:

“Correspondingly, in cells, the blunting of growing microtubule plus-ends by Vinblastine was correlated with reduced EB1 targeting.”

6) Putting the biochemistry all together. It seems that the authors have on-rate constants (directly measured + modeled), as well as off-rate constants (from lifetimes in their single molecule assays on various lattices), which yields K_D values. They also have equilibrium data on tips, closed lattices, disrupted lattices, etc., at specific EB concentrations. Do these numbers all line up?

We have added a new supplemental Figure 3F to predict the average ratio of EB1-GFP equilibrium binding to closed microtubule structures vs disrupted microtubule structures (Figure 2C), based on theoretical binding curves generated from the on and off rate constants measured in this study. In all cases, the predicted values are similar to the experimental equilibrium binding measurements in Figure 2C-E.

7) Killer experiment. From reviewer #1I don't know if it's fair to ask, but there is an obvious killer experiment. The author's argument is that EB1 has a higher on-rate to incomplete sites + a higher off-rate. But because the on-rate is proportionally higher than the off-rate, they get an increase in binding at equilibrium (Figure 2C). This argument implies that EB will flux more through the incomplete sites, even if the steady-state occupancy is higher. So, they should be able to test that directly with FRAP. E.g., take the disrupted vs. closed lattices, FRAP them, and observe faster recovery to a higher equilibrium fluorescence.

Given that FRAP fluorescence recovery is controlled by the off-rate of molecules from the microtubule, we estimate that with a dwell time of ~20 ms, the FRAP recovery half-time would be ~15 ms. Thus, we attempted to use very rapid imaging and a fast, high-powered laser to perform this experiment, however, were not able to photobleach effectively. We concluded that while this is a very interesting experiment, it will require careful planning, and we are currently investigating a new FRAP laser for our laboratory. Therefore, we view this as a longer-term experiment, and will continue our efforts in this regard as future work.